# Sunlight-driven simultaneous $CO_2$ reduction and water oxidation using indium-organic framework heterostructures

Zhongjie Cai[1], Hongwei Liu [2], Jiajun Dai[3], Bao Li[1], Liming Yang [1], Jingyu Wang [1] ✉ & Huaiyong Zhu [4]

Overall artificial photosynthesis, as a promising approach for sunlight-driven $CO_2$ recycling, requires photocatalysts with efficient light adsorption and separate active sites for coupling with $H_2O$ oxidation. Here we show a In-based metal–organic framework (MOF) heterostructure, i.e., In-porphyrin (In-TCPP) nanosheets enveloping an In-$NH_2$-MIL-68 (M68N) core, via a facile one-pot synthesis that utilises competitive nucleation and growth of two organic linkers with In nodes. The coherent interfaces of the core@shell MOFs assure the structural stability of heterostructure, which will function as heterojunctions to facilitate the efficient transfer of photogenerated charge for overall photosynthesis. The In-TCPP shell in MOFs heterostructure improves $CO_2$ adsorption capabilities and visible light absorption to enhance the photocatalytic $CO_2$ reduction. Simultaneously, In-O sites in M68N core efficiently catalyze $H_2O$ oxidation, achieving high yields of HCOOH (397.5 µmol $g^{-1}$ $h^{-1}$) and $H_2O_2$ (321.2 µmol $g^{-1}$ $h^{-1}$) under focused sunlight irradiation. The superior performance of this heterostructure in overall photosynthesis, coupled with its straightforward synthesis, shows great potential for mitigating carbon emissions and producing valuable chemicals using solar energy.

The simultaneous conversion of $CO_2$ and $H_2O$ into HCOOH and $H_2O_2$ through photocatalysis represents an innovative approach that mirrors natural photosynthesis[1,2]. This method has emerged as a key strategy in addressing the escalating concerns of global warming due to $CO_2$ emissions. Recent analyzes have affirmed the economic viability and practical feasibility of formate as a versatile chemical commodity and hydrogen carrier, owing to its adaptability in industrial and energy storage/conversion processes[3–5]. Meanwhile, $H_2O_2$ ranks among the top 100 essential chemicals, commanding a global market share of approximately $4 billion in 2020, with demand projected to reach about $5.2 billion by 2026[6–8]. This underscores the importance of eco-friendly photocatalytic $H_2O_2$ synthesis through water oxidation.

This photocatalytic process, energized solely by solar light, presents an opportunity to exploit sunlight as a cost-effective and renewable energy source. It can activate robust C=O bonds at near room temperature (298.15 K), facilitated by light generated charges, circumventing the decomposition of $H_2O_2$ and HCOOH observed under elevated temperature. However, the inherent chemical stability of $CO_2$ and the slow oxidation of water necessitate the development of highly effective photocatalysts capable of driving these reactions efficiently[9,10]. Despite numerous studies, the quest for efficient photocatalysts is often hampered by the chemical inertness of $CO_2$ and the sluggish kinetics of water oxidation, limiting the efficacy of many existing systems to partial reactions, often reliant on additional

[1]Key Laboratory of Material Chemistry for Energy Conversion and Storage (Ministry of Education), Hubei Key Laboratory of Material Chemistry and Service Failure, School of Chemistry and Chemical Engineering, Huazhong University of Science and Technology, Wuhan 430074, China. [2]Australian Centre for Microscopy and Microanalysis, University of Sydney, Chippendale, NSW, Australia. [3]Institute of Chemistry and Biochemistry, Freie Universität Berlin, Arnimallee 22, Berlin, Germany. [4]School of Chemistry and Physics, Queensland University of Technology, Brisbane, QLD, Australia. ✉ e-mail: wangjingyu@hust.edu.cn

photosensitizers and sacrificial agents (Table 1). MOFs, crystalline solids characterized by uniform pores, large surface area, tunable chemical structure, and semiconductor characteristics, stand out as promising candidates for gas capture and catalysis[11–14]. MOFs are assembled through the interconnection of metal ions with organic ligands, providing a versatile platform for integrating light-harvesting units and metal catalytic sites. The coordination units and organic linkers can efficiently harness light energy as light antennas to induce the generation of charge carriers, which are subsequently transferred to catalytic sites to promote reactions[15–17]. Even so, single MOF structures have limited overall photosynthesis effectiveness (Table 1)[18,19]. Therefore, specific reductive and oxidative units need to be included in hybrid frameworks to enhance their photocatalytic activity, as illustrated by a few representative examples in Table 1. Nevertheless, creating such hybrid structures poses significant synthetic challenges, including complex, multistep synthesis and the need for precise organic ligands, often resulting in only moderate catalytic performance. In the context of the photocatalytic process involving coupled $H_2O$ oxidation and $CO_2$ reduction, efficient transfer of photogenerated charges and absorbed light energy between the incorporated component and MOF structure is essential. These transfers demand precise energy alignment and a well-matched connection between the component and MOF. Moreover, achieving catalyst stability requires a solid combination of the incorporated component and MOF structure. Finally, for the industrial application of catalytic overall photosynthesis, it is crucial to harvest sufficient solar energy in the visible light range[10].

The selective conversion of $CO_2$ to HCOOH using In-based MOF catalysts has been extensively explored due to their strong adsorption of key intermediates, such as COOH* species, and the suppression of competing hydrogen evolution reactions[20–22]. However, single-structure In-based MOFs often suffer from limited catalytic activity of overall photosynthesis, primarily due to the quenching of photogenerated charges and difficulties in integrating distinct active sites required for both half-reactions. To address these challenges, we constructed a heterostructure comprising two MOF structures with light-harvesting units for broad visible light response and distinct metal catalytic sites for reduction and oxidation reactions. The difference in growth kinetics of indium (In) with two types of organic linkers enables a one-pot approach that reduces the synthesis complexity and facilitates the formation of a coherent interface between the two MOF lattices by sharing the same metal atoms. This design leverages unique organic linkers to establish built-in electric fields and atomic-scale charge-transfer pathways, thereby promoting directed charge separation and enhancing overall catalytic performance under visible-light irradiation without relying on external photosensitizers, noble metals, or sacrificial agents (Table 1 and Supplementary Table 1).

## Results

### One-pot synthesis of core-shell M68N@In-TCPP heterostructure

In this study, two popular linkers, tetrakis(4-carboxyphenyl) porphyrin (TCPP) and 2-amino-1,4-benzenedicarboxylic acid (NH₂-BDC), along with indium nitrate were employed for the one-pot synthesis. This process yielded a heterostructure comprising In-porphyrin nanosheets (In-TCPP) enveloping another indium-containing MOF, In-NH₂-MIL-68 (M68N). Zeta potential measurements revealed electrostatic attraction between the oppositely charged In-TCPP and M68N, indicating a strong interactive relationship (Supplementary Fig. 1). The rapid nucleation of NH₂-BDC with $InO_4(OH)_2$ within the first 30 min led to the formation of M68N with uniform rod-like morphology, averaging 10 μm in length (Supplementary Fig. 2), and displaying characteristic hexagonal facets (Fig. 1a). In contrast, In-TCPP exhibits a bulk stacking layers structure due to strong π–π stacking interaction among its metalloporphyrin complexes[23,24] (Supplementary Fig. 3), and its crystallization occurred more slowly. This difference in

crystallization kinetics between the two MOFs enabled the envelopment of M68N by In-TCPP nanosheets (Supplementary Fig. 4a–h). The core@shell architecture of this heterostructure can be seen in Fig. 1b and schematically depicted in Fig. 1c, which also highlights the secondary building units and morphological characteristics of the component MOFs.

The one-step approach yielded M68N@In-TCPP heterostructures with varying compositions. The optimal M68N@In-TCPP heterostructure, demonstrating optimal photocatalytic performance, was obtained at a molar ratio of TCPP:NH₂-BDC as 1:9. Scanning electron microscope (SEM) images and element mapping are provided in Fig. 1d–e and Supplementary Fig. 5. Transmission electron microscopy (TEM) images and the accompanying schematic in Fig. 1f–j depict the structural evolution of the heterostructure, showing how In-TCPP sheets gradually thicken on the stable M68N core. The diffraction patterns maintain the characteristic PXRD peaks for both MOFs within the M68N@In-TCPP heterostructure (Fig. 1k, Supplementary Fig. 6), affirming the preservation of their crystalline features[25]. Both M68N and In-TCPP are orthogonal structures, with changes in the PXRD patterns underscoring the growth kinetics of the heterostructure. Figure 1l shows the rapid nucleation and inherent structural stability of the M68N core, evidenced by the swift appearance of its distinctive peaks within 10 min of reaction. In contrast, the peak corresponding to In-TCPP at 7.5° emerged after 1 hour, indicating slower nucleation. The consistent PXRD peaks confirm the successful integration within the M68N@In-TCPP heterostructure. Increasing the molar ratio of TCPP ligand amplifies the dominance of the crystal structure of In-TCPP, weakening the diffraction signals of M68N in the heterostructures (Supplementary Fig. 7). The disappearance of the (001) diffraction peak of In-TCPP signals its transition from bulk to nanosheet structural form[21], with weak signals from In-TCPP attributed to low TCPP content and competitive nucleation with M68N affecting its crystallinity[26].

Fourier transform infrared (FT-IR) and ${}^1$H nuclear magnetic resonance (${}^1$H NMR) spectroscopy analyzes corroborate the successful synthesis of the heterostructure without the formation of new chemical bonds between two linkers, as detailed in Supplementary Figs. 8–11. The X-ray photoelectron spectrometer (XPS) spectra of In $3d$ revealed a shift to lower binding energy in M68N@In-TCPP, indicating an average valence state of indium below +3 ($In^{σ+}$, $2 < σ < 3$) (Fig. 1m)[20,21,27]. Initially, both the indium nodes and In-N sites exhibit a +3 valence state (Figs. 1m and Supplementary Fig. 12). Due to the faster nucleation kinetics of the core M68N compared to In-TCPP, indium remains in the +3 oxidation state during the first 0.5 hours before gradually shifting to lower binding energies. The TCPP linker competitively coordinates with unsaturated surface indium nodes (In(III)-O-In(II)), increasing the In(II)/(In(III) + In(II)) ratio and resulting in an average indium oxidation state between +2 and +3. When the TCPP (In-N) content is further increased to 75%, the In 3d spectra shift back close to the +3 state (Supplementary Figs. 12 and Table 2). In single-component MOFs, indium mainly exists as +3 when coordinating with each organic linker. During the self-assembly process of In-TCPP, the kinetics of In coordinating with carboxyl group is much faster than that with porphyrin center. Consequently, most of In atoms were coordinated with carboxylic groups of TCPP to form a bulk stacking MOF structure due to strong π–π stacking interaction, while a little part of In atoms were coordinated with porphyrin center to yield only 30% of metalation porphyrin with the coexistence of more uncoordinated pyrrolic N atoms (Supplementary Fig. 11). During the formation of MOF heterostructures, indium species exist in a +3 oxidation state within the first 0.5 h and then gradually shift to a lower binding energy. The main peak at 531.5 eV in O $1s$ XPS spectra of M68N and In-TCPP is assigned to the C=O of the carboxyl group that is connected with In nodes (Fig. 1n)[20]. The O $1s$ peak shifts to higher binding energy (531.8 eV) with extended reaction time. The difference in growth kinetics between the two MOFs enabled the envelopment of

**Table 1 | Performance comparison of known MOF/COF-based catalysts for overall photosynthesis and half $CO_2$ reduction reaction**

| Photocatalyst | Exepriment conditions | Reduction product ($\mu$mol g⁻¹ h⁻¹) | Oxidation product ($\mu$mol g⁻¹ h⁻¹) | Add SA and/or PSᵃ | AQY (%) | Ref. |
|---|---|---|---|---|---|---|
| Fe-In-TCP (Porphyrin-based MOF) | 300 W Xe lamp (>400 nm) $CO_2$, $H_2O$ | HCOOH (17.6) | $H_2O_2$ (13.04) | - | n.r. | 59 |
| MCOF-Ti₆Cu₃ | 300 W Xe lamp $CO_2$, $H_2O$ (> 400 nm, 0.4 W cm⁻²) | HCOOH (169.0) | $O_2$ (n.r.ᵇ) | - | n.r. | 31 |
| NNU-31-Zn (Zn-based COFᶜ) | 300 W Xe lamp (> 400 nm, 0.4 W cm⁻²) $CO_2$, $H_2O$ | HCOOH (26.3) | $O_2$ (12.6) | - | 0.035 (420 nm, 29 mW cm⁻²) | 67 |
| Bi-TTCOF-Zn | 300 W Xe lamp (> 420 nm) $CO_2$, $H_2O$ | CO (11.6) | $O_2$ (5.8) | - | n.r. | 68 |
| PCN-601 (Ni-based Porphyrin MOF) | 300 W Xe lamp (>410 nm, 0.25 W cm⁻²) $CO_2$, $H_2O$ | CO (6.0) CH₄ (10.1) | $H_2O_2$ (37.5) | - | 0.064 (405 nm, 64 mW cm⁻²) | 62 |
| Eu-bpy-Ru-CuCl₂ᵈ | 300 W Xe lamp (> 420 nm, 0.15 W cm⁻²) $CO_2$, MeCN/$H_2O$ | n.dᵉ | n.d | HCOOH (304) | n.r. | 69 |
| COF-367-Coᴵᴵᴵ (Co-based Porphyrin COF) | 300 W Xe lamp (> 380 nm) $CO_2$, MeCN/$H_2O$ | n.r | n.r | HCOOH (93) | n.r. | 70 |
| UiO67-Ir-Cou 6/Cuᶠ | 300 W Xe lamp (> 420 nm, 0.2 W cm⁻²) $CO_2$, MeCN/$H_2O$ | n.d | n.d | HCOO⁻ (408) | n.r. | 71 |
| Ru(phen)₃-Eu-MOF | 300 W Xe lamp (> 420 nm) $CO_2$, MeCN/$H_2O$ | n.d. | n.d | HCOO⁻ (960) | n.r. | 72 |
| MOF-808-EDTA | 300 W Xe lamp (> 420 nm, 1.58 W cm⁻²) $CO_2$, MeCN/$H_2O$ | n.d | n.d. | HCOOH (167) | n.r. | 49 |
| M68N@In-TCPP | 300 W Xe lamp (> 400 nm, 0.3 W cm⁻²) Sunlightᵍ (1.3 W cm⁻²) $CO_2$, $H_2O$ | HCOOH (121.1) CO (22.3) HCOOH (397.5) CO (61.2) | $H_2O_2$ (119.3) $O_2$ (5.9) $H_2O_2$ (321.2) $O_2$ (not detected) | - | 0.16 (420 nm, 9.6 mW cm⁻²) | This work |

ᵃSA sacrificial agents, PS photosensitizers; ᵇnr not reported; ᶜCOF covalent organic framework, ᵈEu-bpydc, bpydc = 2,2'-bipyridine-5,5'-dicarboxylate, integrate with Ru(bpy)₃ photosensitizer (PS); ᵉnd not detect; ᶠUiO67-Ir-Cou 6: UiO-67 MOF integrate with Ir-ppy and coumarin 6 PS; ᵍSunlight: The sunlight intensity was gathered and enhanced by the condenser.

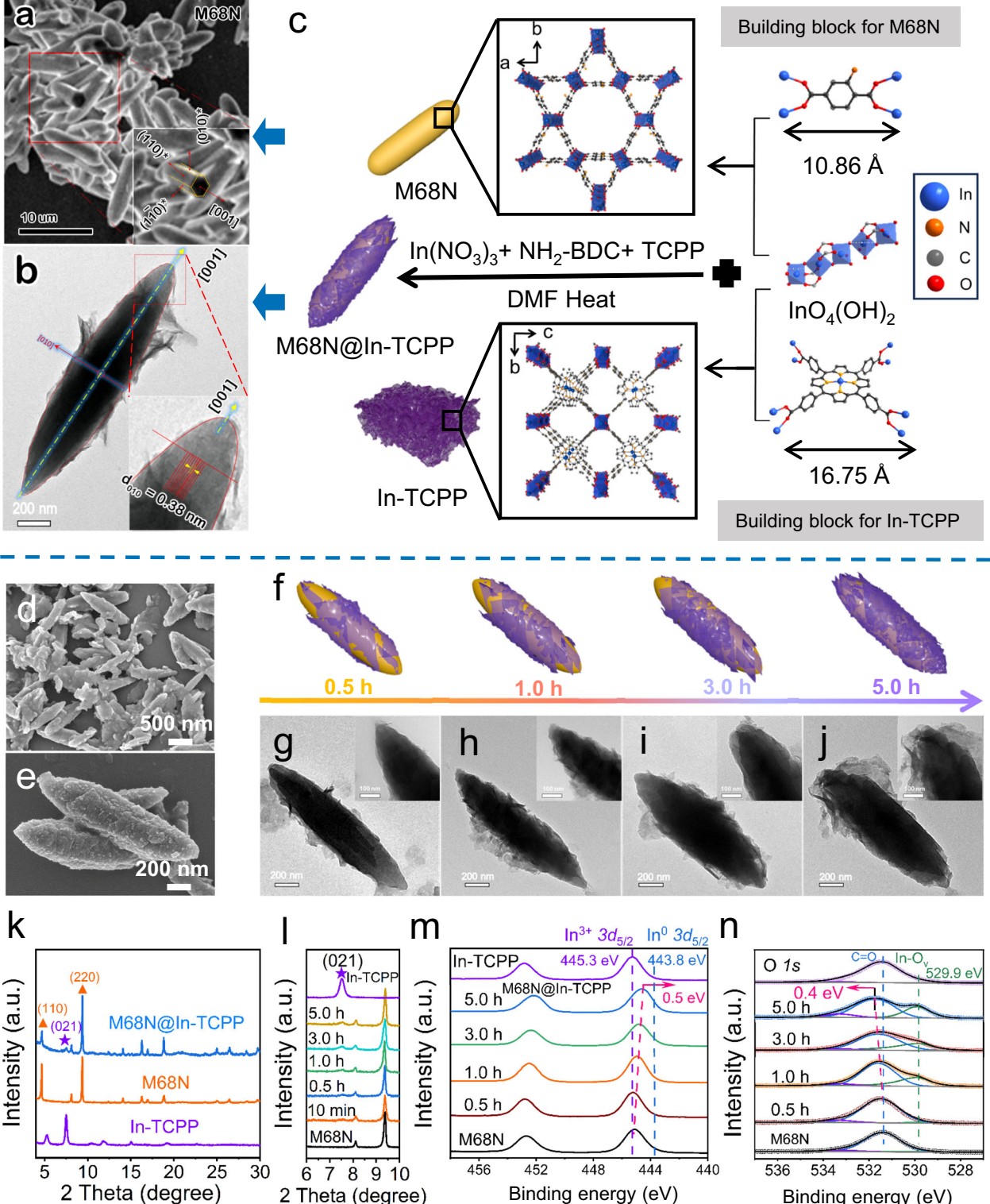

**Fig. 1 | One-pot synthesis and characterizations of core-shell M68N@In-TCPP heterostructure. a** SEM image showing a cluster of NH₂-MIL-68(M68N) nanorods. The area highlighted by a hollow red square represents the region of interest (ROI), showing an exposed cross-section of a nanorod. The inset highlights the pseudo-hexagonal shape of the cross-section. **b** TEM bright field image of an isolated M68N nanorod in the heterostructure at 30 min, displaying excellent mirror symmetry along the growth direction. The inset is a magnified view of the ROI, corresponding to the area marked by a hollow red square. **c** Building block size and crystal structure of M68N and In-TCPP. Schematic representation for producing M68N, In-

TCPP, and M68N@In-TCPP core@shell MOFs structures via the one-pot method. **d, e** SEM images of M68N@In-TCPP. **f** Schematic models and their corresponding TEM images of M68N@In-TCPP nanostructures with extended reaction time. The model depicted the structural evolution with the increase of In-TCPP sheet on the outer of the M68N@In-TCPP heterostructure. **g–j** TEM images of M68N@In-TCPP at various reaction times (Insert showing enlarged images). **k, l** PXRD patterns of samples and M68N@In-TCPP formation. XPS spectra of (**m**) In 3*d* and (**n**) O 1*s* of M68N@In-TCPP at different reaction time. Source data are provided as a Source Data file.

M68N by In-TCPP nanosheets with more exposed In sites due to their slow growth kinetics. In return, a new shoulder peak at around 529.9 eV, corresponding to the lattice oxygen vacancies of In-O ($O_v$), becomes more pronounced. With extended reaction time, the molar ratio of indium to oxygen atoms on the surface increased from 0.16 to 0.38 (Supplementary Table 3), indicating the gradual formation of In-TCPP nanosheets enveloping the M68N core. These results demonstrate that the formation of defective indium sites with oxygen vacancies is essential for the heterostructure formation, arising from the unsaturated indium nodes at the interface between the two MOFs and In-TCPP nanosheets. The N $1s$ XPS spectrum of M68N can be deconvoluted into two peaks located at 397.8 and 399.6 eV, corresponding respectively to C–N and N–H of $NH_2$-BDC ligand (Supplementary Fig. 13)[28]. Within In-TCPP, the two main peaks observed at 397.8 eV and 400.1 eV are assigned to uncoordinated N and pyrrolic N from the porphyrin center, respectively. The additional peak at 398.9 eV of M68N@In-TCPP is likely attributed to N atoms bound to indium sites, accompanied by a 0.3 eV upshift due to the formation of an In-N bond[29], which becomes pronounced with an increase in reaction time. These results affirm the anisotropic growth of In-TCPP on the surface of M68N. The increased ratios of In/N atoms during the growth process support the formation of heterostructures with defective In metal sites[30]. Surface wettability measurements, shown in Supplementary Fig. 8b, further support the presence of the In-TCPP shell in the heterostructure. M68N displays a significantly lower water contact angle (26.6°) than In-TCPP (85.8°) due to the hydrophilicity of -$NH_2$ groups. The water contact angle of the M68N@In-TCPP heterostructure is 69.8°, aligning closer to that of In-TCPP and confirming that the In-TCPP layer effectively covers the In-TCPP layer the surface of the heterostructure.

## Sunlight-driven coupled $CO_2$ reduction and $H_2O$ oxidation

To evaluate the practical feasibility of using sustainable sunlight, we conducted outdoor experiments with a solar concentrator (Fig. 2a) between January 10 and 13th, 2024. This setup provided light intensities varying from 1.0 to 1.3 W cm$^{-2}$. Notably, the experiment did not require organic solvents, sacrificial agents, or additional photosensitizers and cocatalysts (Supplementary Fig. 14). Gas and liquid products were monitored using gas chromatography (GC), ion chromatography (IC), and $^1$H NMR[31]. The product yields and the solar-to-chemical (STC) conversion efficiency ($\eta$ %) demonstrated a correlation with sunlight intensity (Fig. 2a). The highest yield of HCOOH was 397.5 µmol g$^{-1}$ h$^{-1}$, accompanied by a carbon monoxide (CO) yield of 61.2 µmol g$^{-1}$ h$^{-1}$. The $H_2O_2$ yield was slightly lower at 321.2 µmol g$^{-1}$ h$^{-1}$ compared to the HCOOH yield, potentially due to its decomposition under intense light conditions. The $O_2$ yields were not reported due to the difficulty in accurately detecting trace amounts of $O_2$ in the sunlight-driven $CO_2$ photoreduction system using offline injection methods. Additional tests under unfocused natural sunlight (75 mW cm$^{-2}$) showed comparable performance to the xenon lamp with an AM 1.5 G filter, confirming the reliability of the sunlight-driven photocatalysis system (Supplementary Fig. 15). Despite noticeable fluctuations in light intensity (1.0–1.3 W cm$^{-2}$), the average $\eta$% value can be calculated to be 0.04 ± 0.003 %, surpassing the performance of most recently reported $CO_2$ photoreduction systems (Supplementary Table 4). The high product yields achieved under intermittent sunlight irradiation conditions highlight the efficiency of our photocatalysis system in performing overall photosynthesis and demonstrate its significant potential for practical sunlight-driven photocatalysis.

## High yield of overall photosynthesis with the heterostructure photocatalyst

Indoor experiments on overall photosynthesis were conducted under visible light irradiation in $CO_2$-saturated pure water, using an online test setup to explore photocatalytic activity (Supplementary Fig. 16).

Product evolution rates were determined through standard curves (Supplementary Figs. 17–19), which showed negligible product yields in the absence of light for the three MOFs (Supplementary Fig. 20). Among these, the M68N@In-TCPP heterostructure outperforms the other MOFs in $CO_2$ reduction under a constant light intensity (0.3 W cm$^{-2}$), achieving a HCOOH evolution rate of 121.1 µmol g$^{-1}$ h$^{-1}$ significantly higher than the rates for M68N (16.7 µmol g$^{-1}$ h$^{-1}$) and In-TCPP (15.8 µmol g$^{-1}$ h$^{-1}$) as shown in Fig. 2b and Supplementary Fig. 21. The limited activity of M68N and In-TCPP can be attributed to the lack of electron transfer from the ligand to metal nodes, a result of insufficient overlap between the empty $d$ orbitals of In$^{III}$ and the π orbital of its ligands. The enhanced efficiency of the M68N@In-TCPP heterostructure is due to the enhanced charge separation by the interface between two MOFs, effective $CO_2$ adsorption by the microporous In-TCPP shell, and the activation of $CO_2$ at the surface In$^{o+}$ defect sites[32]. Importantly, the system did not show detectable $H_2$ evolution, indicating a preferential $CO_2$ reduction over $H_2O$ reduction, attributed to the higher overpotential required for $H_2$ evolution reaction (HER) compared to $CO_2$ reduction over In-based catalysts. This result can be verified by the linear sweep voltammetry curves of M68N@In-TCPP in Supplementary Fig. 22. Strong adsorption of *COOH intermediates at In sites also helps suppress the competing HER, as corroborated by the previously reported low HER activity work[20,21]. In the absence of $CO_2$, only small amounts of $H_2$ and $H_2O_2$ were detected in argon (Ar) atmosphere, confirming the unfavorable $H_2O$ reduction process on M68N@In-TCPP (Supplementary Fig. 23). The control experiments with the additional introduction of $H_2O_2$ or HCOOH indicate the preferential reduction of $CO_2$ to HCOOH over $H_2O_2$ or $H_2O$ reduction, and the preferential oxidation of $H_2O$ over HCOOH or $H_2O_2$ oxidation (Supplementary Figs. 24, 25, and Supplementary Table 5)[20,21].

In water oxidation, the optimized heterostructure exhibited average production rates of 119.3 µmol g$^{-1}$ h$^{-1}$ for $H_2O_2$ and 5.9 µmol g$^{-1}$ h$^{-1}$ for $O_2$. The electrons generated through $H_2O$ oxidation are closely matched to those consumed during $CO_2$ reduction, confirming efficient overall photosynthesis. The high selectivity for HCOOH (85.4%) and $H_2O_2$ (94.8%) during the reaction aligns with minimal production of byproducts like CO and $O_2$, surpassing the performance of previously reported photocatalysts for overall photosynthesis (Supplementary Table 1). In contrast, a mechanical mixture of M68N and In-TCPP (M68N+In-TCPP), mirroring the content of the heterostructure, exhibited photocatalytic performance similar to M68N alone (Fig. 2b). The absence of M68N resulted in almost no $H_2O_2$ production over In-TCPP. Further exploration using related MOFs—$NH_2$-UiO-66 (Zr) with the same $NH_2$-BDC ligand and In-carboxylate framework (CPM-5 with In node)—confirmed that neither could catalyze the oxidation of $H_2O$ to $H_2O_2$[33–35], as shown in Supplementary Fig. 26. These results underline M68N's crucial role in water oxidation with the heterostructure system.

Control experiments in an Ar atmosphere revealed that $H_2O$ oxidation over the heterostructure photocatalyst occurs independently of $CO_2$ reduction (Fig. 2b), as indicated by comparable $H_2O_2$ production rates between M68N and M68N+In-TCPP. This indicates that $H_2O$ oxidation is closely linked to electron consumption by $CO_2$ at the reductive sites of In-TCPP. The e$^-$/h$^+$ ratio in Fig. 2b is 1.09, which slightly deviates from the theoretical value due to the formation of undetected reactive oxygen species and the decomposition of $H_2O_2$ (Supplementary Fig. 24). This observation aligns with many reported works[16]. By adding an aqueous solution of $H_2O_2$ and HCOOH, negligible decomposition of $H_2O_2$ and HCOOH was observed after the reaction (Supplementary Fig. 27), demonstrating that the liquid products $H_2O_2$ and HCOOH can remain stable and coexist in the system. Regarding $CO_2$ reduction, In-TCPP alone demonstrates a high selectivity for HCOOH production (81%), close to that of the heterostructure and higher than M68N (65%), underscoring the In-TCPP nanosheet shell's beneficial role in $CO_2$ adsorption and reduction. Meanwhile, the

                                                                          

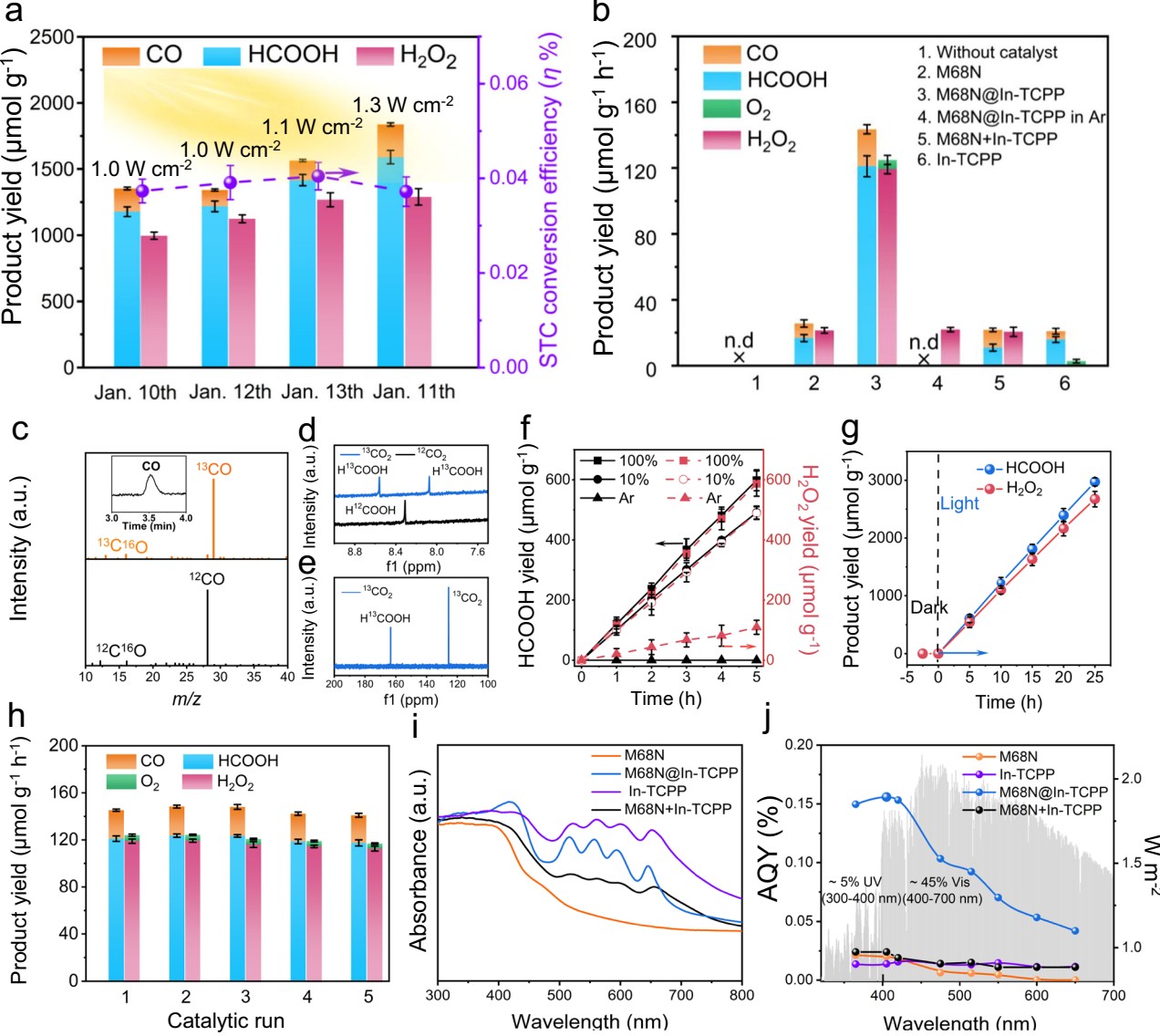

**Fig. 2 | High performance of sunlight-driven photoreduction CO$_2$ and coupling H$_2$O oxidation. a** Practical performance test and corresponding STC conversion efficiency of M68N@In-TCPP photocatalyst under 4 h of natural sunlight irradiation between January 10 and 13th, 2024. **b** Product yields over different photocatalysts under visible light ($\lambda \geq 400$ nm). **c** GC-MS result of $^{13}$CO, **d** $^1$H NMR, and **e** $^{13}$C NMR spectra of liquid products from photocatalytic CO$_2$ reduction using $^{12}$CO$_2$ or $^{13}$CO$_2$. **f** Photocatalytic performance of M68N@In-TCPP in diluted CO$_2$ (10 vol% CO$_2$ in Ar).

**g** Long-term measurement of HCOOH and H$_2$O$_2$ production from overall photo-catalytic CO$_2$ reduction. **h** Products from photocatalytic reaction during 5-cycle tests. **i** UV-vis DRS spectra of different samples. **j** AQY values under various wavelengths of light irradiation. The overlap is the spectrum of solar irradiation. The experimental error bars represent the standard deviations of three independent measurements. Source data are provided as a Source Data file.

hydrophilic M68N provides effective sites for H$_2$O adsorption and oxidation to H$_2$O$_2$. The pH-dependent experiment reveals that M68N@In-TCPP exhibits optimal HCOOH and H$_2$O$_2$ yields at pH = 7 (Supplementary Fig. 28). The production rates of HCOOH and H$_2$O$_2$ remarkably decreased in the reaction system with a high concentration of H$^+$ or OH$^-$ (pH = 3 or 12). The CO$_2$ reduction involves two protons and two-electrons transfer process. The HCOOH formation is limited by the poor CO$_2$ solubility in water at lower pH and is almost suppressed by the inefficient proton supply at higher pH. It is noted that the net consumptions of H$^+$ and OH$^-$ in the overall reactions are zero. The H$_2$O$_2$ production involving H$_2$O as proton donor is also slowed by the high concentration of H$^+$ or OH$^-$[36]. The similar pH-dependent trend in each half-reaction suggests the simultaneous CO$_2$ reduction and water oxidation on M68N@In-TCPP photocatalysts. Additional experiments using sacrificial donors (Supplementary Fig. 29) confirm that H$_2$O as the electron donor achieves higher selectivity toward

HCOOH, thus illustrating the advantages of the M68N@In-TCPP heterostructure in the overall CO$_2$ photoreduction reaction. The origin of carbon-containing products was validated through a $^{13}$C isotopic labelling experiment, analyzed through gas chromatography-mass spectrometry (GC-MS) and $^{13}$C NMR analyzes. In the chromatogram, the 3.51-minute peak corresponding to the CO product exhibited a dominant ion fragment peak at $m/z = 29$, detected after the photocatalytic reaction in a $^{13}$CO$_2$ atmosphere (Fig. 2c), confirming the carbon source of CO product from CO$_2$. Liquid products were analyzed by $^1$H NMR and $^{13}$C NMR. In $^{12}$CO$_2$ atmosphere, the produced H$^{12}$COOH signal displays a singlet peak at 8.30 ppm in $^1$H NMR spectrum. When replacing $^{12}$CO$_2$ by $^{13}$CO$_2$, HCOOH products are detected as doublet peaks at 8.56 and 8.07 ppm arising from $^1$H-$^{13}$C $J$-coupling (Fig. 2d)[7,37,38], whereas the H$^{12}$COOH signal is negligible due to the high $^{13}$CO$_2$ purity of 99.9%. In addition, $^{13}$C NMR spectrum showed a distinct signal for $^{13}$HCOOH at 163.4 ppm (Fig. 2e). The results indicate that the formation

of HCOOH was exclusively originated from $CO_2$ reduction. The porous In-TCPP layer's efficacy in $CO_2$ adsorption and reduction benefits the MOF heterostructure's potential for practical application. The photocatalytic performance was also evaluated using simulated flue gas compositions with $CO_2$ concentrations of approximately 10-15%[29,39]. Experiments with a 10:90 $CO_2$/Ar mixture demonstrated that the M68N@In-TCPP heterostructure exhibit nearly the same performance as with pure $CO_2$, producing 99.85 μmol $g^{-1}$ $h^{-1}$ of HCOOH and 96.77 μmol $g^{-1}$ $h^{-1}$ of $H_2O_2$ (Fig. 2f).

To assess the stability of the heterostructure frameworks, duration and recycling experiments were conducted. The M68N@In-TCPP heterostructure maintained consistent activity during a 25-h photocatalytic reaction (Fig. 2g and Supplementary Fig. 30) and showed only a negligible decrease in production rates across five recycling runs, retaining about 91% of its initial activity (Fig. 2h). Thermogravimetric analysis (TGA) demonstrated that M68N@In-TCPP remains stable up to 350 °C, exhibiting less weight loss than M68N, suggesting that the In-TCPP covering inhibits the decomposition of surface functional group in M68N (Supplementary Fig. 31). The intact crystal structure of each MOF in the heterostructure, confirmed by comparing PXRD patterns of fresh and used samples (Supplementary Fig. 32), and the uniform core@shell morphology of the recycled M68N@In-TCPP (Supplementary Fig. 33), along with consistent FT-IR and XPS spectra (Supplementary Figs. 34 and 35), further confirm the chemical stability. The reaction solution was detected by inductively coupled plasma mass spectrometry (ICP-MS) and $^1$H-NMR. There are no characterized signals of TCPP or $NH_2$-BDC fragments after reactions and only less than 0.19 wt % of the metal nodes are leaching after long-term reaction (Supplementary Fig. 36 and Table 6). To further rule out the possibility that the MOF's organic linkers serve as proton donors, we conducted control experiments by replacing $H_2O$ with $CH_3CN$. Under these conditions, no detectable HCOOH and no signals attributable to the linker confirm that the organic framework does not contribute protons (Supplementary Fig. 37). These findings indicate that the MOF framework remains stable, with no detectable release of linker fragments, and that $H_2O$ is the sole proton source. The robustness of In-based MOFs, derived from strong In-carboxyl coordination bonds and coherent MOF–MOF interfaces, ensures efficient and stable overall photosynthesis[15,40].

We further investigate the light-dependent product yields over the heterostructure. UV-visible diffuse reflectance spectra (UV-vis DRS) presented in Fig. 2i reveal that all MOFs under study exhibit substantial light absorption capabilities across a broad visible spectrum. In-TCPP demonstrates enhanced light absorption compared to M68N, attributed to the light-harvesting capabilities of the porphyrin unit. This includes absorption in the Soret-band (S-band) in the 400-450 nm region, due to π-π* transition, and the Q-band, characterized by four peaks between 450 and 700 nm, resulting from n-π* transitions[41]. The incorporation of the In-TCPP layer atop the M68N surface endows the heterostructure with the characteristic absorption bands, with notably higher intensity than both the mechanical mixture and pure M68N (Fig. 2i). This is particularly evident in the augmented absorption of the S-band, with a prominent peak around 420 nm, indicating more exposed porphyrin units within the outer layer of sheet-like In-TCPP[41].

Consequently, the apparent quantum yield (AQY) of the M68N@In-TCPP heterostructure at 420 nm was measured to be 0.16% with light intensity of 9.6 mW $cm^{-2}$ (Fig. 2j, Supplementary Table 7), significantly exceeding the individual contributions of M68N (0.019%) and In-TCPP (0.014%). The AQY of the physical mixture of M68N and In-TCPP (M68N+In-TCPP) at 420 nm is 0.015%, only a tenth of the core@shell M68N@In-TCPP heterostructure. This highlights the heterostructure's superior photocatalytic performance and demonstrates the practical feasibility of using sustainable sunlight with light intensities ranging from 0.3 to 1.3 W $cm^{-2}$, surpassing those used in most recently reported $CO_2$ photoreduction systems (Table 1 and Supplementary Table 1). Notably, AQY trends align closely with changes in light absorption across the samples (Supplementary Fig. 38 and Supplementary Table 7). While M68N displayed minimal CO and HCOOH production at wavelengths beyond 480 nm, the M68N@In-TCPP heterostructure showed substantial AQYs across the wavelength range of 400-650 nm, aligning with the high-intensity region of sunlight (Fig. 2j). Furthermore, the AQY values for the heterostructure across the entire visible spectrum notably exceed those of the single-component MOFs. This underscores the M68N@In-TCPP heterostructure's suitability for harnessing the wavelength distribution of solar light for efficient photocatalytic activity.

## Mechanism studies of coupling photocatalytic $CO_2$ reduction and $H_2O$ oxidation

The surface property and reactant adsorption of the catalysts were examined to further identify the catalytic sites for $CO_2$ reduction and $H_2O$ oxidation. All MOFs exhibit a type-I $N_2$ adsorption-desorption isotherms, indicating a structure with abundant micropores (Supplementary Fig. 39)[42]. Based on the Brunauer–Emmett–Teller (BET) model, the specific surface area value ($S_{BET}$) of In-TCPP was calculated to be 1107 $m^2$ $g^{-1}$, significantly exceeding that of M68N (Supplementary Table 8). Wrapping the M68N core with In-TCPP nanosheets increased the specific surface area, micropore volume, and affinity to $CO_2$. The In-TCPP shell formation notably enhances the surface microporous structure of M68N@In-TCPP, which, despite only a 10% molar ratio of In-TCPP, shows a doubled $CO_2$ uptake capacity of 21.8 $cm^3$$g^{-1}$ at 298 K and 1.0 bar, compared to 10.2 $cm^3$$g^{-1}$ for M68N (Fig. 3a). This enhanced $CO_2$ adsorption is attributed to the large π-conjugated structure of In-TCPP, enriched with polarizing pyrrole groups that have a strong affinity for $CO_2$[29,43].

Surface wettability analysis highlighted a greater affinity of $H_2O$ towards M68N, as evidenced by its lower water contact angle (26.6°) compared to 85.8° for In-TCPP, which is ascribed to the hydrophilic -$NH_2$ groups present in M68N (Supplementary Fig. 8b). This high affinity facilitates the $H_2O$ oxidation reaction on M68N. TGA analysis of $H_2O$-saturated samples unveiled distinct desorption behaviors: M68N exhibited a two-stage desorption pattern, with water desorbing initially below 90 °C and subsequently at elevated temperature ranging from 160 to 200 °C[44]. In contrast, water desorption from In-TCPP proceeded more readily, reaching a plateau at -90 °C, indicating a weaker interaction with $H_2O$ (Supplementary Fig. 40). However, the desorption from M68N@In-TCPP required higher temperature (-160 °C), suggesting the pivotal role of M68N in $H_2O$ adsorption within the heterostructure. Density function theory (DFT) calculations provided a comprehensive analysis of the adsorption performance of $CO_2$ and $H_2O$ on $In_2O_4(OH)_2$ and In-N sites, including optimal adsorption energies and bond lengths (Supplementary Data 1). Figure 3b, c display the reactant adsorption as $CO_2$ at In-N site and $H_2O$ at $InO_4(OH)_2$ sites, as compared to the reversed adsorption models in Supplementary Fig. 41. These calculations suggest that $H_2O$ is more readily activated at $InO_4(OH)_2$ sites. In contrast, $CO_2$ adsorption is favored at In-N sites with a lower adsorption energy and a shorter bonding distance with the O atom of $CO_2$ (Supplementary Fig. 42). The comparison in the electron density redistribution supports the transfer of photoinduced electrons from In-TCPP to $CO_2$ at In-N sites and from the adsorbed $H_2O$ to $InO_4(OH)_2$ sites[45].

Electron paramagnetic resonance (EPR) spectra were used to further identify catalytic sites and their interactions with reactants via in-situ diffuse reflectance Fourier transform infrared spectroscopy (DRIFTS). The EPR spectrum of M68N showed no significant signals (Fig. 3d), suggesting a lack of paramagnetic centers under the tested conditions. In contrast, the EPR spectrum of In-TCPP displayed distinct signals at $g$ = 1.988, 2.005, 2.015, attributed to the $In^{III}$ node, the delocalized π electron of the porphyrin ring, and the doublet ($S$ = 1/2) $In^{III}N_4$ complex, respectively, with $g$ values around $g_{xx}$ = $g_{yy}$ = 2.005,

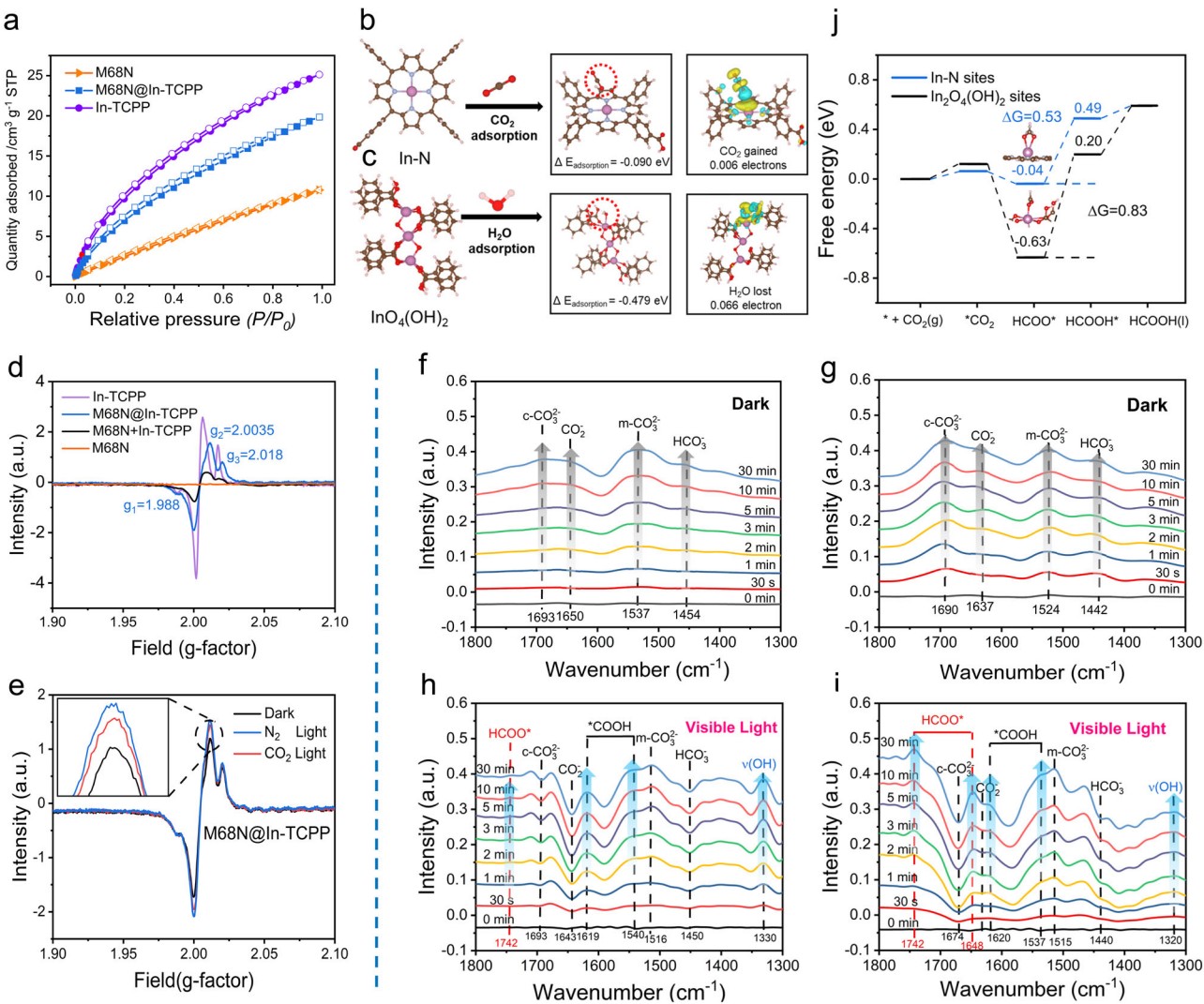

**Fig. 3 | Detection of reaction active sites and mechanism. a** The $CO_2$ adsorption-desorption isotherms of samples up to 1.00 bar at 298 K. **b** DFT calculations on the charge difference density of $CO_2$ adsorption at In-N sites. In: magenta, O: red, N: light blue, C: brown, H: pink. **c** DFT calculations on the charge difference density of $H_2O$ adsorption at $InO_4(OH)_2$ sites. **d** EPR spectra of M68N, In-TCPP, M68N@In-TCPP, and mechanical mixture of M68N and In-TCPP (M68N+In-TCPP) under visible light ($\lambda \geq 400$ nm). **e** Changes in the EPR spectra of M68N@In-TCPP before and after light irradiation in $N_2$ and $CO_2$ atmosphere. The insert is the magnified area of the selected region. In situ DRIFTS spectra from 1300 to 1800 $cm^{-1}$. Reactants adsorption on **f** M68N and **g** M68N@In-TCPP in the dark. Intermediates evolution on **h** M68N and **i** M68N@In-TCPP during photocatalytic reaction. **j** Calculated Gibbs free energy profile for the production of HCOOH on In-N and $In_2O_4(OH)_2$ sites of M68N@In-TCPP. Source data are provided as a Source Data file.

$g_{zz} = 2.015$ (Fig. 3d and Supplementary Fig. 43). These findings align with previously reported on metalloporphyrin materials[46,47]. In the EPR spectrum of M68N@In-TCPP heterostructure, we observed a shift in the porphyrin signal to $g = 2.0035$, indicative of the formation of oxygen vacancies at the defective In-O sites[48]. In contrast, the EPR signal for a mechanical mixture of M68N and In-TCPP, although at the same $g$ factor as In-TCPP, exhibited significantly lower intensity. The absence of defective In sites in the mechanical mixture suggests that the formation of heterostructure creates the defective sites. Upon exposure to light, the M68N@In-TCPP exhibited an increased peak intensity at $g = 2.0035$ (Fig. 3e), indicating light-driven electron transfer from the porphyrin unit. During the photocatalytic reaction, the defective In sites showed a slight increase in intensity within the first 5 hours and then remained stable up to 20 hours (Supplementary Fig. 44). Additionally, the principal $g$-values of these sites did not change, confirming the stable structure and valence state of In in $In_2O_4(OH)_2$. These findings illustrate that the defective In sites remain stable during extended operation, ensuring sustained photocatalytic performance. When we replaced $N_2$ with $CO_2$, the signal intensity diminished, reflecting electron consumption by adsorbed $CO_2$ molecules[49,50], providing valuable insight into the dynamic electron transfer mechanisms that drive the catalytic activity under light irradiation.

Figure 3f-i present an in-situ DRIFTS study comparing $CO_2$ adsorption and activation with and without a porous In-TCPP layer. The observed changes in the spectra under dark conditions highlight the adsorption states of $CO_2$ and $H_2O$ on the catalyst's surface. On the M68N surface, adsorbed $CO_2$ molecules predominantly exist as $HCO_3^-$ (1454 $cm^{-1}$), monodentate carbonated (m-$CO_3^{2-}$, 1537 $cm^{-1}$), $CO_2^-$ radical (1650 $cm^{-1}$), and chelating-bridged carbonate (c-$CO_3^{2-}$, 1693 $cm^{-1}$) (Fig. 3f)[16,31,51]. The intensified signals of these surface-adsorbed carbon species in M68N@In-TCPP, as compared to M68N, corroborate a higher affinity for $CO_2$ adsorption (Fig. 3g). Under illumination, the signals corresponding to $HCO_3^-$, m-$CO_3^{2-}$ and c-$CO_3^{2-}$ species gradually diminished, while those of m-$CO_3^{2-}$ signals increased with irradiation time[16,31,51]. Additionally, new peaks corresponding to *COOH (1540,

1619 cm$^{-1}$) and HCOO* (1742 cm$^{-1}$), intensified with prolonged irradiation (Fig. 3h and Supplementary Table 9). This suggests photocatalytic conversion of adsorbed HCO$_3^-$ and c-CO$_3^{2-}$ species on the M68N surface to CO$_2^-$, and subsequently to HCOO* and *COOH, the crucial intermediate of HCOOH and CO[19]. During the photocatalytic reaction on M68N@In-TCPP surface, the bands of HCO$_3^-$ and c-CO$_3^{2-}$ bands depleted, while the *COOH signal increased compared to that on M68N surface (Fig. 3i). Furthermore, additional peaks at 1648 and 1742 cm$^{-1}$, corresponding to HCOO*, further underscore the efficient consumption of adsorbed carbonate, facilitating the production of HCOOH and *COOH (1620 and 1537 cm$^{-1}$)[31,51]. The hydroxyl stretching vibrations in the range of 3500–3800 cm$^{-1}$ indicate the adsorption of H$_2$O on both M68N and M68N@In-TCPP (Supplementary Fig. 45)[52,53]. During the photocatalytic process, a strong absorption peak at 1320 cm$^{-1}$ arises from the O–H deformation vibration of the adsorbed H$_2$O$_2$ product[16]. EPR with 5,5-dimethyl-1-pyrroline N-oxide (DMPO) is used to study the possible generation of •OH radicals accompanied by H$_2$O$_2$ formation. In the absence of photocatalyst, there were no peaks corresponding to •OH in the EPR spectrum. During the photocatalytic reaction on M68N@In-TCPP, a distinct 1:2:2:1 quadruplet peak in EPR spectrum is attributed to the DMPO–•OH adducts (Supplementary Fig. 46). Since the HOMO of M68N (1.8 V vs NHE) is not high enough to oxidize water to produce •OH radicals (H$_2$O + h$^+$ = •OH + H$^+$, 1.97 V vs NHE), the •OH radicals can be generated by the reduction of H$_2$O$_2$ on photocatalyst surface (H$_2$O$_2$ + e$^-$ = •OH + OH$^-$, 0.73 V vs NHE)[54–56]. Even the produced •OH can react with H$_2$O$_2$ via the equation (H$_2$O$_2$ + •OH = •O$_2^-$ + H$_2$O + H$^+$), the consumption of H$_2$O$_2$ can be avoided due to its low dosage as well as the rapid release into the solution[57,58]. To explore the mechanism of water oxidation in CO$_2$ atmosphere, we conducted the photocatalytic experiments with different scavengers. Tert-butanol (TBA) and KBrO$_3$ were introduced as the scavenges for •OH radicals and electrons, respectively (Supplementary Fig. 47). The addition of TBA slightly decreased the H$_2$O$_2$ production (112.2 µmol g$^{-1}$ h$^{-1}$), excluding the possibility of an •OH-mediated single-electron pathway for water oxidation. In contrast, the production rate of H$_2$O$_2$ was significantly increased from 119.3 to 185.4 µmol g$^{-1}$ h$^{-1}$ in the presence of KBrO$_3$. The accelerated H$_2$O-to-H$_2$O$_2$ oxidation rate can be ascribed to the increased charge separation by rapid consumption of electrons. Further, the change of •OH concentration was monitored by a fluorescence probe of 2-hydroxyterephthalic acid and compare to that of H$_2$O$_2$ concentration during the photocatalytic reaction (Supplementary Fig. 48). Different from the linear increase in H$_2$O$_2$ production, the •OH generation increased at the initial 30 min of reaction and gradually became saturated with the prolonged light irradiation. The distinct tend between them indicates that H$_2$O$_2$ should not be formed by •OH coupling. Instead, the saturation state of •OH concentration resembles the feature as its formation from H$_2$O$_2$ reduction[57]. It can be concluded that photocatalytic H$_2$O oxidation in CO$_2$ generates H$_2$O$_2$ over M68N@In-TCPP via a two-electron oxidation process. Thus, the M68N@In-TCPP heterostructure enhances the adsorption and conversion of CO$_2$ and H$_2$O through plausible pathways of CO$_2$ → CO$_2^-$ → HCOO* → HCOOH and H$_2$O → H$_2$O$_2$[16,59]. Based on the Gibbs free energy of reaction intermediates, the CO$_2$ reduction pathway involves two hydrogenation steps: from *CO$_2$ to HCOO* and from HCOO* to HCOOH*, with the latter being the rate-determining step (Figs. 3j, Supplementary Data 1). This lower energy barrier at In-N sites than InO$_4$(OH)$_2$ sites indicates a more favorable pathway for HCOOH formation, thereby establishing In-N as the active site for CO$_2$ photoreduction.

## Efficient charge transfer within M68N@In-TCPP

The coupling of CO$_2$ reduction at In-TCPP and H$_2$O oxidation at M68N within the M68N@In-TCPP heterostructure necessitates charge transfer between the two MOFs, which depends on energy alignment and coherent interfaces between the MOFs. Energy bandgaps were

determined using Tauc plots from UV-vis DRS spectra, showing that M68N has a band gap of 2.73 eV (Supplementary Fig. 49a), while In-TCPP has discrete energy levels with band gaps ranging from 1.73 to 1.95 eV (Supplementary Fig. 49b). The flat band potentials, corresponding to the lowest unoccupied molecular orbital (LUMO) levels of the MOFs, were inferred from Mott-Schottky plots[12], revealing LUMO energy levels of −0.93 and −0.88 V versus the normal hydrogen electrode (NHE) for M68N and In-TCPP, respectively (Supplementary Fig. 49c, d). These values enabled calculation of the highest occupied molecular orbital (HOMO) levels at 1.8 and 0.85 V versus NHE for M68N and In-TCPP, respectively, indicating their suitability to facilitate CO$_2$ reduction to HCOOH.

Additional insights were gained from transformed Kubelka-Munk function plots and DFT calculations on the band structure, revealing that In-TCPP's band gap is narrower than that of M68N (Supplementary Fig. 50). Density of states calculations indicated that In-carboxylic chains connected to metalloporphyrin significantly influence the HOMO of In-TCPP (Supplementary Fig. 51). The results suggest that the CO$_2$ reduction is likely facilitated at the metalloporphyrin sites, with M68N showing a superior capability for H$_2$O oxidation due to its more positive HOMO energy relative to In-TCPP. Ultraviolet photoelectron spectroscopy (UPS) was used for identifying the energy levels of HOMO within these MOFs (Supplementary Fig. 52). The work function ($W_F$) values were determined to be 4.0 eV for M68N and 3.75 eV for In-TCPP. This difference suggests that In-TCPP has a relatively higher Fermi level than M68N prior to interaction (Supplementary Fig. S53a). The formation of the M68N@In-TCPP heterostructure creates an interfacial electric field (IEF), which facilitates electron transfer from In-TCPP to M68N until the Fermi levels equilibrate (Supplementary Fig. 53b). This IEF results in an upward bending of energy levels in In-TCPP and downward bending in M68N, establishing heterojunctions between M68N and In-TCPP interfaces that enhance charge separation and transfer[60]. Compared to pristine M68N and the mechanical mixture, the heterostructure demonstrates a significant enhancement in absorption within the 400 nm to 650 nm wavelength range, alongside noticeable shifts in absorption peak wavelengths (Supplementary Fig. 54). The M68N@In-TCPP heterojunctions, formed by the interface between the two MOFs, mimic a reaction pathway similar to natural photosynthesis, where sunlight drives the conversion of CO$_2$ and H$_2$O into carbohydrates in green plant leaves.

The interfacial boundary between the core nanorod M68N and the attached shell nanoflake In-TCPP is clearly demonstrated by the morphological evolution from faceted polyhedral to curve-edge nanorod of M68N, as shown in SEM and TEM. This morphological transformation indicates a strong interfacial binding due to the attachment of In-TCPP nanoflakes, which significantly alters the orientation anisotropy and surface energy of M68N, consequently changing its Wulff morphology. Electronic diffraction patterns (EDPs) were obtained under a low-Miller index zone axis perpendicular to the growth axis. These patterns helped identify the crystallographic features influencing the growth morphology of M68N. TEM bright-field images show the annotated facets (100) and (140), with the projected facets parallel to the growth direction (Supplementary Fig. 55a–c). The corresponding EDPs for these facets were designated as [100] and [-22, 5, 16] crystallographic orientations, as seen in Supplementary Fig. 55b–d. A composite stereographic projection centered at [001] in Supplementary Fig. 55e illustrates the alignment parallel to the common axis between the two parallel planes. The SEM image in Fig. 1a shows the well-crystallized symmetrical morphology of the M68N nanorod and the cross-section perpendicular to the growth direction. In the region of interest (ROI), a fractured nanorod exposes the cross-section with a pseudo-hexagonal shape enveloped by three sets of flat planes. This morphology is consistent with the TEM bright field image shown in Fig. 1b, where lattice fringes parallel to the long axis are visible in the magnified image of the top corner, with a lattice space of

3.8 nm assigned to the (010) plane parallel to the growth direction [001].

Based on these detailed observations, a schematic 3D morphology of the nanorod is constructed (Fig. 4a), indicating exposed facets of (010), (110) and (-110). This schematic allows for the reconstruction of the surface energy envelope, enabling the use of a Wulff plot for further morphological analysis of the cross-section. In Fig. 4b, each point on the pseudo-hexagonal cross-section profile represents a crystalline direction or a plane normal, with facets (010), (110) and (-110) having very low surface energy, located at the surface energy singularity points. This detailed examination underscores the profound impact of interfacial interactions on the structural and morphological properties of the M68N nanorod. Supplementary Fig. 56a shows a high-resolution TEM (HRTEM) image of an interface area between M68N and In-TCPP. The In-TCPP nanoflakes are clearly visible, adopting the same polygonal shape. Two specific regions within this image, marked with hollow red squares, are further magnified in Supplementary Fig. 56b, d. The lattice fringes in these regions were converted into fast Fourier transform (FFT) patterns, which are then used for diffraction indexing. In Supplementary Figs. 56c, e, the two FFT patterns derived from the lattice fringes are assigned to the zone axes [510] and [100], respectively. The exposed facets of the nanoflake are indexed as (021) and (0-21), indicating the nanoflakes project as a square.

Both MOFs exhibit body-centered orthogonal lattices with similar lattice parameters. The presence of Bain strain necessitates a minor angle rotation to minimize lattice distortion following their mutual growth. According to the invariant deformation element (IDE) model for diffusional phase transformation[61], the shortest vector in M68N should remain non-rotated after growth. The actual rotation angle is approximately 2.4°, closely matching the initial Bain lattice correspondence. Based on this orientation relationship, the plane in M68N that aligns with (100) in In-TCPP is identified as (100) at the interface, and the plane that matches (021) in In-TCPP is (110) in M68N (Fig. 4c). Two potential interface configurations emerge from this analysis: Case I - $(001)_{M68N}//(100)_{In\text{-}TCPP}$ and Case II - $(110)_{M68N}//(021)_{In\text{-}TCPP}$ or $(1\text{-}10)_{M68N}//(0\text{-}21)_{In\text{-}TCPP}$. These configurations are visually represented by 3D atomic models in Fig. 4d, e, showing how the In-O octahedra are shared between M68N and In-TCPP at the interface boundary, essential for the effective integration and performance of the heterostructure.

The charge carrier dynamics between M68N and In-TCPP, facilitated by their interface, are further elucidated through the analysis of both steady-state photoluminescence (PL) spectra and time-resolved photoluminescence (TRPL) spectra. M68N displays a broad emission band peaking at 475 nm within the visible light range, indicative of its active photophysical properties (Supplementary Fig. 57). In contrast, In-TCPP shows distinct metalloporphyrin emissions, particularly the S-band and Q-band emissions at approximately 650 and 710 nm, respectively[41]. The Q-band emission is more intense than the S-band, suggesting a self-quenching phenomenon via $S_2$-to-$S_1$ energy transfer, which amplifies the intensity of the Q-band emission.

Further insights come from the TRPL decay spectra, which disclose intricate charge carrier dynamics within M68N and In-TCPP (Fig. 4f, g, Supplementary Table 10). When selectively excited at 375 nm, M68N shows an average charge carrier lifetime ($\tau_{Ave}$) of 0.78 ns (Fig. 4f). This lifetime extends significantly to 4.39 ns within the M68N@In-TCPP heterostructure, indicating more efficient charge transfer processes at the material interface. The S-band emission decay of In-TCPP follows a mono-exponential profile with a lifetime of 1.81 ns, reducing to 0.71 ns within M68N@In-TCPP (Fig. 4g). Conversely, the Q-band emission presents a two-component decay profile with a lifetime of 2.55 ns ($\tau_1 = 14.85$, 0.01%; $\tau_2 = 2.55$, 99.99%), where the longer $\tau_1$ indicates the photoinduced charge transport and the shorter $\tau_2$ corresponds to porphyrin-localized exciton recombination

(Supplementary Fig. 58)[62]. The exceptionally small percentage (0.01%) of decay exhibiting longer $\tau_1$ reflects the rapid electron-hole recombination in In-TCPP. In stark contrast, the Q-band emission of M68N@In-TCPP exhibits a bi-exponential decay characterized by $\tau_1 = 0.69$ ns (56.5 %) and $\tau_2 = 1.47$ ns (43.5%). The presence of a substantial fraction of decay with a longer $\tau_2$ indicates enhanced charge separation capabilities of the heterostructure[63]. To probe the charge carrier dynamics in real-time, ultrafast transient absorption (TA) spectroscopy was used (Fig. 4h). Kinetic fitting of GSB recovery reveals that the average lifetime within the heterostructure decreases from 482 ps to 288 ps (Fig. 4i, Supplementary Table 11). In contrast, the physical mixture, M68N+In-TCPP, has an average lifetime of 430 ps—comparable to pure In-TCPP (484 ps)—but much longer than that of M68N@In-TCPP (Supplementary Fig. 59). Concurrently, the SE process lifetime in M68N@In-TCPP (66 ps) shows a considerably faster decay than that of In-TCPP alone (494 ps) and M68N+In-TCPP (449 ps). This significant reduction in the lifetimes corroborates the more efficient charge transfer through the interface of two MOFs in the heterojunction compared to physical mixture[64–66].

Integrating the findings from TA and TRPL analyzes, it becomes clear that the formation of M68N@In-TCPP heterojunctions significantly enhances charge separation and transfer, crucial for improving the efficiency of coupled $CO_2$ reduction and water oxidation. This enhancement is further substantiated by electrochemical and photoelectrochemical analyzes. Electrochemical impedance spectrum (EIS) results for the M68N@In-TCPP display the smallest semicircle radius in the high-frequency region, indicating that the interfaces between M68N and In-TCPP substantially increase electron transfer efficiency (Supplementary Fig. 60). Additionally, transient photocurrent measurements indicate that the photocurrent intensity of M68N@In-TCPP is considerably stronger than that of either MOF alone or the M68N+In-TCPP mixture under visible light (Fig. 4j). Remarkably, in a $CO_2$ atmosphere, the photocurrent of the heterostructure is twentyfold compared to that in an argon atmosphere (Supplementary Fig. 61), demonstrating efficient electron transfer to adsorbed $CO_2$ molecules. Together, these results highlight that the enhanced charge transfer in M68N@In-TCPP arises from the coherent interface between M68N and In-TCPP, reinforcing the advantages of the core@shell heterostructure.

Figure 5 presents a tentative mechanism for the coupled photocatalytic conversion of $CO_2$ and $H_2O$ into useful chemicals, driven by charge transfer processes between M68N and In-TCPP. As depicted in Fig. 5a, electrons in the HOMOs of M68N and In-TCPP are excited to their LUMOs under light irradiation. According to the energy level alignment, the LUMOs of both M68N and In-TCPP are positioned at more negative than the reduction potential needed for converting $CO_2$ to HCOOH. The electrons excited to the LUMO of M68N are transferred across the interface between M68N and In-TCPP to the HOMO of In-TCPP, where they recombine with holes, thus facilitating effective charge separation and transfer. Interestingly, only the HOMO level of M68N is more positive than the potential necessary for the oxidation of $H_2O$ to $H_2O_2$, which enables this oxidation reaction. Although the photocatalytic performance slightly declines with an increase in irradiation wavelength, the AQY value of the M68N@In-TCPP heterostructure remains significantly higher than that of the individual MOFs, highlighting the critical role of the heterojunctions in facilitating efficient charge transfer (Fig. 5b). Figure 5c succinctly illustrates the dynamic process of charge transfer and the roles of catalytic sites during the photocatalytic reactions under visible light irradiation. In this process, the oxidation of adsorbed $H_2O$ molecules to $H_2O_2$ primarily occurs at the $InO_4(OH)_2$ sites of M68N via a two-electron oxidation pathway. Concurrently, electrons are transferred from M68N to In-TCPP. At In-TCPP, $CO_2$ molecules, which are preferentially adsorbed on the In-N sites, are reduced to HCOOH through a two-electron

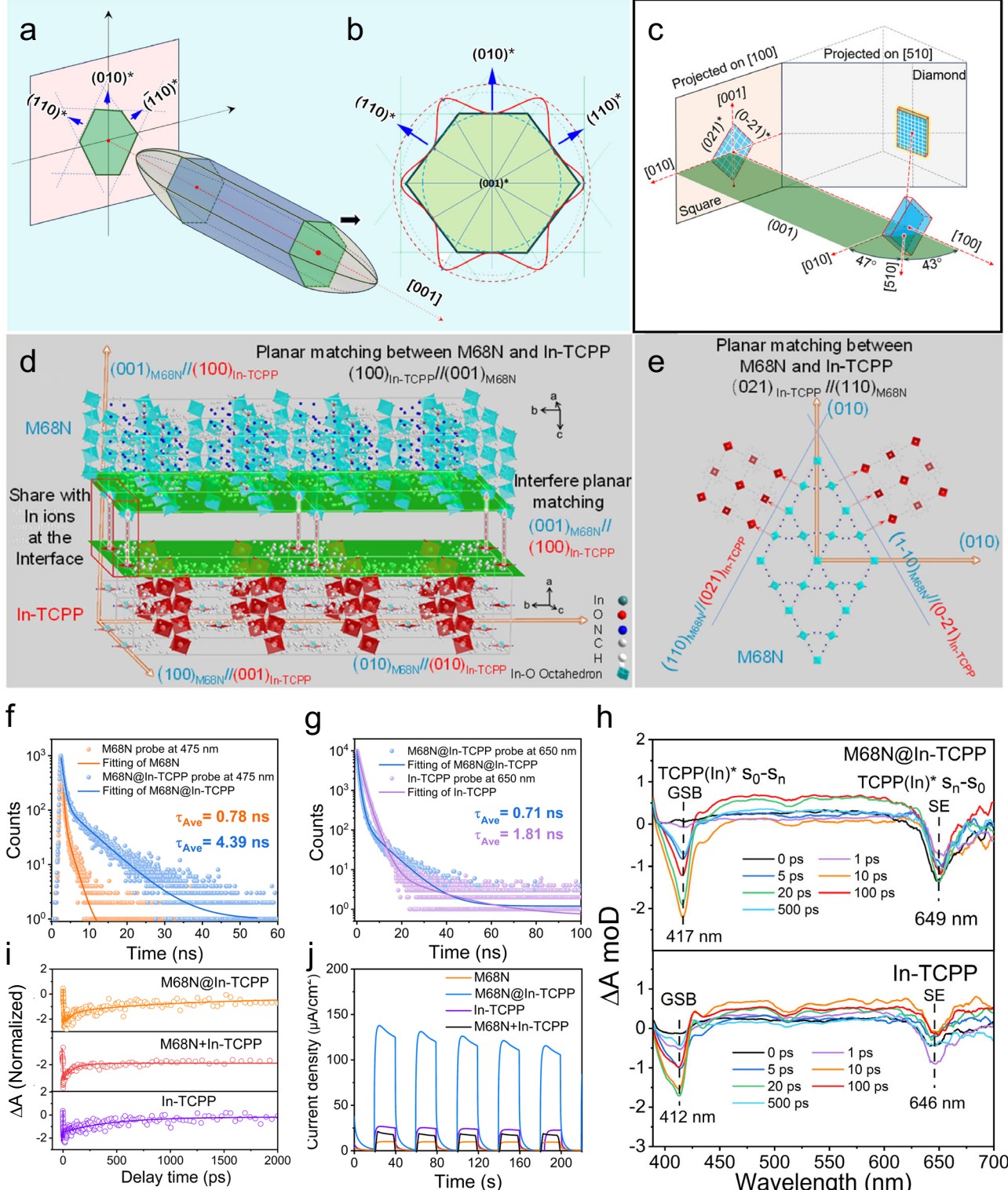

**Fig. 4 | Charge transfer mechanism of stable interface probed by transient spectra. a** A 3D schematic of the M68N nanorod morphology, bordered by two types of flat facets, (110) and (010), with a curved facet at each corner of the double-head cone. **b** An estimation of the surface energy envelope and the Wulff plot of the cross-section perpendicular to the growth direction [001], based on the observed geometric features. Determination of the morphology of In-TCPP mixed with M68N. **c** the reconstructed 3D morphology of the In-TCPP nanoflake. It looks like a square pattern in the projected direction [100] and a diamond shape along [510].

3D atomic models of the possible interface configuration between M68N nanorod and In-TCPP nanoflake. **d** Case I, $(001)_{M68N}$ // $(100)_{In-TCPP}$. **e** Case II, $(110)_{M68N}$ // $(021)_{In-TCPP}$. Time-resolved PL decay spectra: **f** Excitation at 375 nm and emission probe delay at 475 nm; **g** Excitation at 485 nm and emission probe delay at 650 nm. **h** Ultrafast transient absorption spectra of In-TCPP and M68N@In-TCPP: pump excitation at 380 nm. **i** TA kinetics curves and their fittings of GSB signals. **j** Transient photocurrent measurements of samples during the light on/off cycle. Source data are provided as a Source Data file.

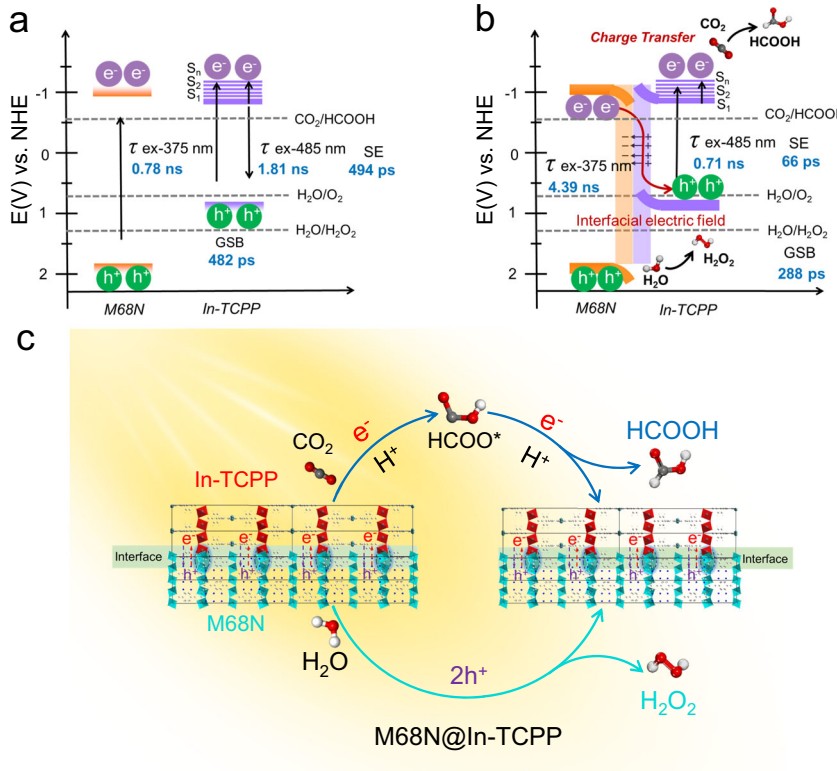

**Fig. 5 | Illustration of charge carrier dynamics and overall photosynthetic pathways at interfaces. a** The band structure of M68N and In-TCPP before contact. **b** The charge transfer process in M68N@In-TCPP via an interfacial electric field upon visible-light irradiation. **c** Schematic representation of the interface structure in the M68N@In-TCPP heterostructure and photocatalytic reaction pathways for coupled $CO_2$ reduction and $H_2O$ oxidation.

reduction pathway. This MOF heterostructure mimics the natural process of photosynthesis, utilizing natural sunlight to achieve high efficiency and promote the sustainable production of chemical fuels by optimising the interplay of charge and energy transfers.

## Discussion

In summary, this research presents a MOF heterostructure composed of two In-based MOFs, notable for its strong visible light absorption, high $CO_2$ affinity, and superior photocatalytic activity for overall photosynthesis under sunlight. This heterostructure was prepared by a simple one-pot synthesis that utilises competitive nucleation and growth of two organic linkers with indium nodes to form a core@shell structure. Within this structure, $CO_2$ reduction predominately occurs at the In-N sites of In-TCPP, while $H_2O$ oxidation takes place at the In nodes of M68N. This system achieves record-breaking yields of HCOOH (397.5 µmol $g^{-1}$ $h^{-1}$) and $H_2O_2$ (321.2 µmol $g^{-1}$ $h^{-1}$) under concentrated sunlight, surpassing the performances of existing MOF and COF-based photocatalysts. The proof-concept study highlights an eco-friendly photocatalytic process, exhibiting high yield (121.1 µmol $g^{-1}$ $h^{-1}$) and selectivity (85.4%) for HCOOH under a broad visible light spectrum (indicate the irradiation light intensity), alongside efficient $H_2O$ oxidation to $H_2O_2$ (119.3 µmol $g^{-1}$ $h^{-1}$). The catalyst maintains stability over five cycles and 25 hours. The high photocatalytic efficiency stems from the synergistic effects of M68N and In-TCPP, which are interconnected by shared In nodes. The interfaces between the two MOFs function as heterojunctions, enhancing charge transfer and separation, significantly boosting the system's efficiency. This investigation not only pioneers an effective approach for designing efficient photocatalysts for $CO_2$ conversion but also showcases the immense potential of MOF heterostructures in harnessing solar energy for environmental remediation and sustainable fuel production.

## Methods

### Materials

All the chemical reagents were purchased from commercial sources and used directly without further purification. Indium nitrate hydrate ($In(NO_3)_3 \cdot xH_2O$, Shanghai Aladdin Biochemical Technology Co., Ltd., 99.99%); 2-amino-1,4-benzenedicarboxylic acid ($NH_2$-$H_2BDC$, Shanghai Energy Chemical Reagent Co., Ltd., 99.99%); 4,4',4'',4'''-(Porphine-5,10,15,20-tetrayl) tetrakis(benzoic acid)($H_2TCPP$, Shanghai Energy Chemical Reagent Co., Ltd., 99 %); Benzoic acid (BA, Shanghai Sinopharm Chemical Reagent Co., Ltd., 99.9 %); N, N-dimethylformamide (DMF, Shanghai Sinopharm Chemical Reagent Co., Ltd., Analytical Reagent); Ethanol (Shanghai Sinopharm Chemical Reagent Co., Ltd., Analytical Reagent); Potassium iodide (KI, Shanghai Aladdin Biochemical Technology Co., Ltd., 99.99%); Potassium biphthalate($C_8H_5KO_4$, Shanghai Aladdin Biochemical Technology Co., Ltd., 99.99%); Deuterium oxide ($D_2O$, Shanghai Energy Chemical Reagent Co., Ltd., 99.99%);Deuterium chloride (DCl, Shanghai Energy Chemical Reagent Co., Ltd., 99.99%); Potassium chloride (KCl, Shanghai Sinopharm Chemical Reagent Co., Ltd., Analytical Reagent); Nafion solution (Sigma-Aldrich, 5 wt%); Ag/AgCl reference electrode and Pt wire electrode were obtained from Gaoss Union. Carbon dioxide gas ($CO_2$, 99.999%), argon gas (Ar, 99.999%), and nitrogen gas ($N_2$, 99.999%) were purchased from Wuhan Zhongxinruiyuan Gas Co., Ltd., The $^{13}CO_2$ was supplied by Guangzhou Yuepujiayuan Special Gases Co., Ltd and the abundance is 99%. Ultra-pure water (resistance > 18.2 MΩ cm) was used in all experimental processes.

### Preparation of catalysts

The synthetic procedure for the preparation of M68N is as follows: $In(NO_3)_3 \cdot xH_2O$ (146 mg, 0.48 mmol), 2-amino-1,4-benzenedicarboxylic acid ($NH_2$-$H_2BDC$) (24 mg, 0.12 mmol), and benzoic acid (BA) (8 mg,

0.06 mmol) were dispersed into a solvent of N, N-dimethylformamide (DMF) (5.0 mL). The above solution was sonicated for 10 min to eliminate air bubbles and then heated to 130 °C and kept for 5.0 h. The light-yellow products were isolated immediately by centrifuging, washing with DMF and ethanol for three times. Finally, it dried under vacuum for 12 h at 60 °C, and the light-yellow crystals of M68N were obtained. In-TCPP was prepared by dispersing $In(NO_3)_3 \cdot xH_2O$ (15 mg, 0.05 mmol) and $H_2TCPP$ (16.0 mg, 0.02 mmol) into DMF (5.0 mL) solvent. Then the mixture was experienced in the similar procedure of M68N to yield purple In-TCPP crystals. For the preparation of M68N@In-TCPP, the experimental procedures are the same as the that of M68N except for adding two organic linkers as 2-amino terephthalic acid ($NH_2$-$H_2BDC$) (16 mg, 0.09 mmol) and $H_2TCPP$ (8.0 mg, 0.01 mmol).

### Measurement of solar-to-chemical conversion efficiency (outdoor natural sunlight)

M68N@In-TCPP photocatalyst was diluted in 15 mL of water and maintained at room temperature (298.15 ± 2 K) via circulating condensate. It was directly irradiated by sunlight in a $CO_2$ atmosphere outdoors on Huazhong University of Science and Technology (HUST) campus (Longitude 114.41, Latitude 30.51), from 12: 00 am to 4:00 pm on January 10 to 13th, 2024 (Supplementary Fig. 14). The sunlight was gathered and enhanced by the condenser to induce photocatalytic $CO_2$ reduction, which provided light intensity with $1.15 \pm 0.15 \, W \, cm^{-2}$ (irradiated area is ca. 3.14 cm²). The solar-to-chemical (STC) conversion efficiency ($\eta\%$) was determined by irradiation under natural sunlight. The reaction equation of photocatalytic overall reduction follows the equation: $CO_2 + 2H_2O \rightarrow HCOOH + H_2O_2$. Therefore, the corresponding STC efficiency was calculated via the following equation:

$$STC = \frac{[\Delta G_{overall}] \times ([\Delta N_{HCOOH}] + [\Delta N_{H_2O_2}])}{I \times S \times T} \times 100\% \qquad (1)$$

The experimental error bars were obtained by the standard deviations of three independent measurements.

### Visible-light driven photocatalytic experiments (indoor online test equipped with Xe lamp)

The experiments were performed in a 350 mL reactor under 1 atm $CO_2$ kept at room temperature (298.15 ± 2 K). The degassed catalyst (5.0 mg) was mixed with 10 mL deionized water and purged with $CO_2$ for 30 min in the reactor. The reactor was stirred by a magnetic stirrer and irradiated by a 300 W Xe lamp at a constant light intensity (0.30 ± 0.05 $W \, cm^{-2}$, irradiated area is ca. 19.62 cm²) with a 400 nm cutoff filter. During the light irradiation, the cool circulated water was used to control the reaction temperature. Photocatalytic gas products were detected by a gas chromatography (GC) analyzer (Agilent−8860, USA) equipped with an online test setup (Supplementary Fig. 16). The corresponding standard curve of $O_2$ is provided in Supplementary Fig. 17. The liquid products were analyzed by ion chromatogram (IC) using a Metrohm 861 basic system (Metrosep A supply e 25- 250/4.0 analytical column), with a detection limit on the order of 10 parts per billion (ppb). Calibration curves were built in the concentration range of 10 μM to 400 μM, which almost covered all concentration ranges of the diluted sample. The actual IC data of HCOOH recorded during the time-dependent photocatalytic $CO_2$ RR process almost all fell within the range of the calibration curve, indicating that the concentrations of HCOOH formed in the reaction systems of concern could be precisely quantified. For the $^1H$ NMR test, 0.5 mL of the reaction solution was sampled and mixed with 0.1 mL of $D_2O$, and adding 0.1 mL of DMSO as internal standard substance. For measuring the apparent Quantum Yield (AQY) of the $CO_2$ photocatalytic reaction process, the Xe lamp equipped with band-pass filters was used as the light source. The corresponding calculation formula is as follows:

$$AQY(\%) = \frac{numbers \ of \ the \ total \ consumed \ electrons}{numbers \ of \ the \ incident \ photos} \times 100\%$$
$$= \frac{(N_{HCOOH} + N_{CO}) \times 2 \times N_A}{P \times S \times \frac{\lambda}{h \times c}} \times 100\% \qquad (2)$$

The $^{13}CO$ isotopic labelling experiment was analyzed by Agilent 7890B gas chromatography-mass spectrometry (GC-MS) with a column of HP-5. An ECZ400S NMR spectrometer analyzed $H^{13}COOH$ isotopic labelling.

The generation of hydrogen peroxide ($H_2O_2$) during the photocatalytic $CO_2$ reduction process was analyzed by an iodimetry method. In detail, 1.0 mL of the product solution was sampled and 0.45 μm of filter membrane was used to filter the catalysts. The transparent reaction solution was obtained for the detection of $H_2O_2$ production. The method of detecting $H_2O_2$ is as follows. In an aqueous solution of 1 mL of 1 M $C_8H_5KO_4$ and 1 mL of 0.4 M KI, $H_2O_2$ reacts with $I^-$ ions to produce $I_3^-$ that has an absorption peak at 350 nm. The UV-vis absorption spectra of the solution were recorded by a UV-vis spectrophotometer (Shimadzu UV-2600), enabling the generated $H_2O_2$ to be quantifically analyzed. The experimental error bars were obtained by the standard deviations of three independent measurements.

## Data availability

The data that support the findings of this study are presented in the paper and Supplementary Information. Atomic coordinates of the optimized models are provided in Supplementary Data. Source data are provided with this paper.

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

## Acknowledgements

J. Wang acknowledges the financial support of the National Natural Science Foundation of China (22122602, 22376071) and the Ministry of Science and Technology of the People's Republic of China (G2023154030L). H. Zhu. acknowledges the financial support from the Australian Research Council (DP210103357). The authors acknowledge the Analysis and Testing Center, Huazhong University of Science and Technology (HUST) for the characterization of materials. The part of computing work in this paper is supported by the Public Service Platform of High Performance Computing by Network and Computing Center of HUST.

## Author contributions

J.W. conceived the project and designed the experiments. Z.C. performed the experiments and analyzed the data. H.L. helped with the structural analysis. J.D. performed the DFT calculations. B.L. discussed the structural results. L.Y. helped with the analysis of DFT calculation. H.Z. helped in mechanism analysis. Z.C., J. W., and H.Z. wrote the manuscript. All of the authors discussed the results and commented on the paper.

## Competing interests

The authors declare no competing interests.
