## [Transparent Peer Review file · Nature Communications]

Sunlight-driven simultaneous CO₂ reduction and water oxidation using indium-organic framework heterostructures

Corresponding Author: Professor Jingyu Wang

Version 0:

Reviewer comments:

Reviewer #1

(Remarks to the Author)

This work presents a novel MOF heterostructure as M68N@In-TCPP via a facile one-pot synthesis that utilises competitive nucleation and growth of two organic linkers with In nodes. The designed MOF heterostructure is interesting as it integrates the reductive In-TCPP sites for CO₂ adsorption and reduction with the oxidative In-O sites for H₂O oxidation using the same In nodes. The coherent interfaces of the core@shell MOFs assure the structural stability of heterostructure, which will function as Z-scheme heterojunctions to efficiently mimic the natural photosynthesis. Therefore, the photocatalytic performance of M68N@In-TCPP is outstanding among the reported results. The results were sufficiently supported by the experiment and theoretical evidences. Although the paper has novelty and well-organized contents, appropriate revisions are required to address the following concerns before it can be considered for publication.

1. There are detailed discussions on the formation and mechanism of Z-scheme heterojunctions in Section of light absorption and energy level. Some descriptions are repeated in the analysis of mechanism related to Fig. 5. Since the charge transfer is further studied by TA and TRPL analyses, the integration of these discussions into the photocatalytic mechanism of Z-scheme heterojunctions is more reasonable.
2. The comparison to the mechanical mixture of M68N and In-TCPP (M68N+In-TCPP) is to clarify the merits of core@shell M68N@In-TCPP heterostructure from the results of visible light absorption (Fig. 2h) and charge transfer (Fig. 3a). The authors should present the AQY values of M68N+In-TCPP versus various wavelengths of light irradiation in Fig. 2i. Meanwhile, the favored charge transfer by the formation of interface needs the evidence on their comparison.
3. The M68N@In-TCPP heterostructure is interesting in terms of overall photosynthesis. The results indicated the In-TCPP sites for CO₂ reduction and the In-O sites for H₂O oxidation. Which one has greater influence on the whole reaction efficiency? The photocatalytic activity of CO₂ reduction should be compared between using H₂O and sacrificial agents as electron donors.
4. The defective In metal sites were formed in the MOF heterostructures to function as active sites for photocatalytic reaction. What about the stability of the defective sites? The EPR spectrum of the catalysts should be monitored to present the changes in oxygen vacancies during long-term test.
5. Although the experiments are thoroughly conducted, the structural stability of the MOF heterostructure needs further study, for example, the effect of H₂O₂ and •OH products on the decomposition of the organic linkers, the leaching of metal sites.

Reviewer #2

(Remarks to the Author)

In this manuscript, the authors report the synthesis of In-based MOF (In-porphyrin (In-TCPP) nanosheets enveloping an In-NH₂-MIL-68 core) for simultaneous photocatalytic CO₂ reduction and H₂O oxidation. They found that the In-TCPP shell enhanced CO₂ adsorption and its subsequent photocatalytic reduction to produce HCOOH, while the In-O sites in the M68N core promoted H₂O oxidation to generate H₂O₂. They attributed the better performance of this heterostructure to the enhanced charge separation and transfer due to the formation of a Z-scheme band structure. However, the significance of this manuscript is minimized by the unclear selectivity and confusable processes of CO₂ reduction and water oxidation. Thus, this paper is not recommended for publication. Some specific comments are as follows.

1. One major concerning point is the selectivity of photocatalytic reduction and oxidation. As we known, H₂O₂ is easier to be reduced than CO₂, while HCOOH can also be oxidized by photogenerated holes. The authors should clearly explain why In-based MOF selectively reduce CO₂ instead of H₂O and H₂O₂, and selectively oxidize H₂O other than HCOOH and

H₂O₂ in more detail.

2. The generation of HCOOH and H₂O₂ should be competitive with water as the only proton source. Although the authors measured the production of HCOOH and H₂O₂ in this core-shell M68N@In-TCPP heterostructure photocatalysis, their quantum yields were very low. Thus, the authors are suggested to further understand the reaction processes.

3. Abstract: The argument that “the inherent catalytic inefficiency of single-structure In-based MOFs” is too general, and the solution of “constructing a heterostructure comprising two MOF structures” for this question is quite rough and simple.

4. As both the CO₂ reduction and H₂O₂ production require protons, the authors should consider the effect of pH on the performance of M68N@In-TCPP.

5. The stability of the material should be evaluated by characterizing the used photocatalysts, as the organic frameworks can serve as the proton donors.

6. Normally a Z-scheme photocatalytic system should have electron shuttles, which are absent in the M68N@In-TCPP. This catalyst is a typical heterojunction, but does not meet the characteristics of Z-scheme photocatalyst.

7. The strong absorption peak at 1266 cm⁻¹ was attributed to the O–H deformation vibration (•OH). Was this •OH a radical or adsorbed OH group? It is very difficult for 1e⁻WOR to form •OH radical.

Reviewer #3

(Remarks to the Author)

This work by Cai et al developed a composite photocatalyst with two MOF nanomaterials based on indium. The formation process of the titled heterostructure is interesting with good elucidation. Moreover, the high affinity between two MOF nanostructures enables synergistic effects in achieving artificial photosynthesis with good yields in CO and H₂O₂. The mechanistic studies also verify the photocatalytic pathways to some extent with rational experimental/computational design. Overall, I recommend the publication of this work on Nature Communications after addressing the following issues.

1. I suggest to put the less important EDX mapping of Figure 1 into SI for higher clarity in presentation.

2. In figure 1m, the XPS of In seems to be very smooth. Did the authors give the smoothed data? Original data points should nonetheless be provided like the XPS O 1s results.

3. The reported record-high activity in CO and H₂O₂ yields compared to other literature is not very telling since the light intensity of the reported system far exceeds one sun (nearly 10 and 40 times). The utilized light intensity and apparent quantum yields should be supplied in Table 1 with abundant discussion on this issue.

4. Why CO and O₂ product yields are absent in Figure 2a and related discussion, while the two products could be detected in another system as shown in Figure 2a and 2g? This is also not consistent with Table 1 where no oxygen yield was given.

5. The detection details of O₂ were not mentioned in Methods.

6. In Figure 3f and 3g, what is the meaning of the 0.2 bars which are absent in Figure 3h and 3i?

7. If the decay was claimed as single-component, there will be no two τ values will be presented. Two-component could be more suitable in discussing Figure S39.

8. I am curious why the produced H₂O₂ would not oxidize HCOOH, two of which are dissolved in water? A discussion needs to be given accordingly.

9. Recent examples on artificial photosynthesis, e.g., *Angew. Chem. Int. Ed.* 2024, 63, e202401344, *J. Am. Chem. Soc.* 2024, 146, 17773, etc., are suggested for citation.

Reviewer #4

(Remarks to the Author)

Reviewer #5

(Remarks to the Author)

An indium-based MOF heterostructure was synthesized using a one-pot method, which serves as an effective photocatalyst for simultaneous CO₂ reduction and water oxidation to yield formic acid (397.5 $\mu\text{mol g}^{-1} \text{h}^{-1}$) and hydrogen peroxide (321.2 $\mu\text{mol g}^{-1} \text{h}^{-1}$), respectively. This manuscript describes a balanced work and is noteworthy. I am happy to recommend this paper for publication in Nature Communications. However, some following concerns need to be addressed:

1. The manuscript synthesized an interesting core-shell M68N@In-TCPP heterostructure through controlling the crystallization kinetics between the two MOFs. However, in Fig.1a and 1b the characteristic hexagonal facets of M68N are challenging to identify. Could you provide a clear picture, such as 3D reconstruction TEM? Moreover, element mapping images don't provide evidence for the optimal M68N@In-TCPP heterostructure (lines 115-116).

2. The synthesis of the MOF heterostructure is described in Figure 1c, why use H₂TCPP instead of In-TCPP? (lines 590-591). In addition, is it possible for the trivalent indium at the center of the porphyrin to coordinate axially? Please provide additional evidence to support it. The details of the ¹H NMR in Supplementary Fig. 11 are unclear, and the calculation detail for porphyrin metalation is difficult to understand (Supplementary lines 220-225).

3. While the shift to lower binding energy in the In 3d spectra suggests a reduction in the average valence state of indium, it would be beneficial to provide a more detailed explanation of the underlying mechanisms contributing to this shift. Specifically, elucidating the factors that lead to the transition of indium species from a +3 state to a lower oxidation state would enhance the understanding of the electronic environment in the heterostructure.

4. The reported increase in the molar ratio of indium to oxygen atoms from 0.16 to 0.38 is noteworthy. XPS measures the

composition of the surface of the material, In node on the surface of the heterogeneous MOF, the ratio of indium to oxygen may theoretically be less than M68N (0.19), but it should not be greater than In-TCPP (0.20). It would strengthen the manuscript to include a discussion on the significance of this increase. For example, how does this change affect the structural integrity and functionality of the heterostructure? The spatial distribution of the MOF heterogeneous interface can be deepened by surface etching and the change of In-O ratio measured by XPS.

5. The expression about the effect of hydrophilicity on the H₂O₂ performance in lines 305-308 is contradictory with the actual performance results.

6. In general, the yield of photocatalysis under natural light is lower than that achieved with xenon lamp irradiation. However, when comparing natural light to visible light irradiation in this manuscript, the yield under natural light can be twice as high (Figure 2a and 2b). Please give the possible discrepancy factors.

7. For the overall photosynthesis, the oxidation and reduction products should have a certain molar ratio according to the reaction equation. In this manuscript, the total number of moles for oxidation products (H₂O₂ and O₂) shown in Figure 2b and 2g is less than the theory, please analyze the reason. In addition, whether the quantification of the product requires multiple tests and displays of error rods.

8. For in-situ DRIFTS spectra of M68N and M68N@In-TCPP in Figure 3g and 3h, what might be the reason for the big difference in the signal-to-noise ratio? And DRIFTS peak is not obvious, whether there is a way to measure more convincing signal strength.

9. The proposed two pathways for CO₂ and H₂O adsorption are intriguing. It would enhance clarity to discuss the physical basis for the preference of CO₂ adsorption at the In-N sites over InO₄(OH)₂ sites in more detail. Additionally, including energy diagrams or transition state structures could help visualize the differences in activation energies and elucidate the underlying mechanisms of the adsorption processes.

Version 1:

Reviewer comments:

Reviewer #1

(Remarks to the Author)

The raised comments have been well addressed and this revised version can be accepted for publication as it is.

Reviewer #2

(Remarks to the Author)

The authors addressed some of my concerns. However, the major concerning point remain unclear, which should be further clarified.

1. Although the authors determined the HOMO and LUMO levels of each component, the HOMO of M68N (1.8 V) is not high enough to oxidize water to produce OH radicals (> 1.9 V).

2. How can the organic material keep stable in the presence of produced OH radicals? Furthermore, OH radicals can also react with H₂O₂ at a high reaction rate.

3. Was H₂O₂ produced via single-electron WOR or two-electron WOR? The authors should discuss the mechanism of H₂O₂ formation with more solid evidences.

4. In the ¹³C₂ isotopic experiment, the signal was very low in comparison with the noise, did authors detect H₁₂COOH and compare it with H₁₃COOH?

5. Minor: The two subtitles "Photocatalytic CO₂ reduction under natural sunlight" and "Photocatalytic CO₂ reduction and H₂O oxidation" in the "Methods" are confusable.

Reviewer #3

(Remarks to the Author)

I can see the enormous efforts from the authors devoted to the revision which meets my requests as well as other reviewers'. Therefore I strongly recommend the publication of this nice work on Nature Communications.

By the way, I agree to remove the claim on Z-scheme which is not the gist of this work.

Reviewer #4

(Remarks to the Author)

Reviewer #5

(Remarks to the Author)

In the revised manuscript, the authors have made a thorough revision following the reviewers' suggestions. The quality and readability of this article are much better than the previous one. Most of the raised concerns have been considered and resolved. Thus, I recommend accepting the manuscript now.

Version 2:

Reviewer comments:

Reviewer #2

(Remarks to the Author)

The authors have addressed all my concerns, I recommend the publication of this manuscript in Nature Communications.

Manuscript NCOMMS-24-54153-T Response to Reviewers' Comments

Given below are our responses (in BLUE colour) to the reviewer' comments. The changes to the manuscript and supplementary information are marked in RED colour.

Reviewer #1

Comment. This work presents a novel MOF heterostructure as M68N@In-TCPP via a facile one-pot synthesis that utilises competitive nucleation and growth of two organic linkers with In nodes. The designed MOF heterostructure is interesting as it integrates the reductive In-TCPP sites for CO₂ adsorption and reduction with the oxidative In-O sites for H₂O oxidation using the same In nodes. The coherent interfaces of the core@shell MOFs assure the structural stability of heterostructure, which will function as Z-scheme heterojunctions to efficiently mimic the natural photosynthesis. Therefore, the photocatalytic performance of M68N@In-TCPP is outstanding among the reported results. The results were sufficiently supported by the experiment and theoretical evidences. Although the paper has novelty and well-organized contents, appropriate revisions are required to address the following concerns before it can be considered for publication.

Response: We appreciate the reviewer's detailed summary of our work and are pleased that the novelty and strengths of our MOF heterostructure design and its photocatalytic performance were well recognized. We have carefully addressed each of the reviewer's concerns below.

Comment 1. There are detailed discussions on the formation and mechanism of Z-scheme heterojunctions in Section of light absorption and energy level. Some descriptions are repeated in the analysis of mechanism related to Fig. 5. Since the charge transfer is further studied by TA and TRPL analyses, the integration of these discussions into the photocatalytic mechanism of Z-scheme heterojunctions is more reasonable.

Response: Thanks for your suggestion. We have integrated the discussions on charge transfer into the photocatalytic mechanism. The photogeneration and transfer of electrons depends on the light absorption and energy level of photocatalyst, so the related descriptions are transferred to the caption of Supplementary Fig. 49 and 52 in supporting information.

We have now reorganized the charge transfer discussions in the photocatalytic mechanism related to Fig.5 on Page 20 line 599: "As depicted in Figure 5a, electrons in the HOMOs of M68N and In-TCPP are excited to their LUMOs under light irradiation. According to the energy level alignment, the LUMOs of both M68N and In-TCPP are positioned at more negative than the reduction potential needed for converting CO₂ to HCOOH. The electrons excited to the LUMO of M68N are transferred across the interface between M68N and In-TCPP to the HOMO of In-TCPP, where they recombine with holes, thus facilitating effective charge separation and transfer. Interestingly, only the HOMO level of M68N is more positive than the potential necessary for the

oxidation of H₂O to H₂O₂, which enables this oxidation reaction.”

Comment 2. The comparison to the mechanical mixture of M68N and In-TCPP (M68N+In-TCPP) is to clarify the merits of core@shell M68N@In-TCPP heterostructure from the results of visible light absorption (Fig. 2h) and charge transfer (Fig. 3a). The authors should present the AQY values of M68N+In-TCPP versus various wavelengths of light irradiation in Fig. 2i. Meanwhile, the favored charge transfer by the formation of interface needs the evidence on their comparison.

Response: Thank you for this valuable suggestion. In response, we have incorporated additional data comparing the mechanical mixture (M68N+In-TCPP) with our M68N@In-TCPP heterostructure. Specifically, we have added the AQY values of M68N+In-TCPP at various wavelengths (Fig. 2i and Supplementary Table 7). At 420 nm, the AQY of M68N+In-TCPP is 0.015%, only one-tenth of that achieved by M68N@In-TCPP (0.16%).

Fig. 2i. AQY values under various wavelengths of light irradiation. The overlap is the spectrum of solar irradiation.

To further support the charge transfer benefits of the heterostructure, we have included time-resolved absorption (TA) spectroscopy of M68N+In-TCPP (Supplementary Fig. 57). Kinetic fitting of ground state bleach (GSB) recovery shows that M68N+In-TCPP has an average lifetime of 430 ps—comparable to pure In-TCPP (484 ps)—but much longer than the 288 ps observed for M68N@In-TCPP. Similarly, the stimulated emission signal at 646 nm for M68N+In-TCPP persists at 449 ps, close to that of In-TCPP alone (494 ps). In contrast, M68N@In-TCPP exhibits a significantly shorter lifetime (66 ps), indicating more efficient charge transfer. Transient photocurrent measurements further confirm that M68N@In-TCPP generates a substantially stronger photocurrent under visible light than either MOF alone or the M68N+In-TCPP mixture (Fig. 4j). Together, these results highlight that the enhanced charge transfer in M68N@In-TCPP arises from the coherent interface between M68N and In-TCPP, reinforcing the advantages of the core@shell heterostructure.

Fig. 4. (i) TA kinetics curves and their fittings of GSB signals. (j) Transient photocurrent measurements of samples during the light on/off cycle.

Supplementary Fig. 57. Ultrafast transient absorption spectra of M68N+In-TCPP: pump excitation at 380 nm.

These findings and discussions have been incorporated into the revised manuscript, where we have added a detailed comparison of AQY values as well as the updated TA spectroscopy and transient photocurrent data.

Page 12 Line 336: “Consequently, the apparent quantum yield (AQY) of the M68N@In-TCPP heterostructure at 420 nm was measured to be 0.16% (Fig. 2i, Supplementary Table 7), significantly exceeding the individual contributions of M68N (0.019%) and In-TCPP (0.014%). The AQY of the physical mixture of M68N and In-TCPP (M68N+In-TCPP) at 420 nm is 0.015%, only a tenth of the core@shell M68N@In-TCPP heterostructure. This highlights the heterostructure's superior photocatalytic performance, ranking it among the top catalysts for overall photosynthesis in a gas-liquid-solid system (Supplementary Table 1).”

Page 19 line 570: “To probe the charge carrier dynamics in real-time, ultrafast transient absorption (TA) spectroscopy was used. Kinetic fitting of GSB recovery reveals that the average lifetime within the heterostructure decreases from 482 ps to 288 ps (Fig. 4i, Supplementary Table 8). In contrast, the physical mixture, M68N+In-TCPP, has an average lifetime of 430 ps—comparable to pure In-TCPP (484 ps)—but much longer than that of M68N@In-TCPP

(Supplementary Fig. 57). Concurrently, the SE process lifetime in M68N@In-TCPP (66 ps) shows a considerably faster decay than that of In-TCPP alone (494 ps) and M68N+In-TCPP (449 ps). This significant reduction in the lifetimes corroborates the more efficient charge transfer through the interface of two MOFs in the heterojunction than physical mixture⁶⁵⁻⁶⁷.

Page 20 Line 585: “Additionally, transient photocurrent measurements indicate that the photocurrent intensity of M68N@In-TCPP is considerably stronger than that of either MOF alone or the M68N+In-TCPP mixture under visible light (Fig. 4j). Remarkably, in a CO₂ atmosphere, the photocurrent of the heterostructure is twentyfold compared to that in an argon atmosphere (Supplementary Fig. 59), demonstrating efficient electron transfer to adsorbed CO₂ molecules. Together, these results highlight that the enhanced charge transfer in M68N@In-TCPP arises from the coherent interface between M68N and In-TCPP, reinforcing the advantages of the core@shell heterostructure.”

Comment 3. The M68N@In-TCPP heterostructure is interesting in terms of overall photosynthesis. The results indicated the In-TCPP sites for CO₂ reduction and the In-O sites for H₂O oxidation. Which one has a greater influence on the whole reaction efficiency? The photocatalytic activity of CO₂ reduction should be compared between using H₂O and sacrificial agents as electron donors.

Response: We appreciate the reviewer’s insightful suggestion and have conducted additional experiments using sacrificial agents as electron donors to replace the H₂O oxidation reaction (Supplementary Fig. 29). Typical sacrificial agents include TEOA, TIPA, and TEA. Compared with overall reaction of CO₂ with H₂O, these sacrificial agents favored the formation of CO while reducing the yield of HCOOH, similar to the literatures’ reports on the selective CO₂-to-CO reduction in the presence of typical sacrificial agents. This result aligns higher energy input required for H₂O oxidation to H₂O₂ (116.7 kJ/mol) as compared to CO₂ reduction to CO (20.1 kJ/mol).

Supplementary Fig. 29. Comparison of electron donors for the photocatalytic performance of M68N@In-TCPP. The sacrificial agents are mixed with CH₃CN (acetonitrile). The volume ratio of sacrificial agent to CH₃CN is 1:4. TEOA: triethanolamine; TIPA: triisopropanolamine; TEA:

triethylamine.

Under photocatalytic conditions, H₂O not only serves as a proton donor but also helps remove photogenerated holes, thereby accumulating separated electrons for CO₂ reduction. Through proton-coupled electron transfer (PCET), a protonated HCOO* intermediate is formed, significantly lowering the activation barriers for both H₂O oxidation and CO₂ reduction (*Nat. Commun.* 2020, 11, 3043; *Angew. Chem. Int. Ed.* 2024, 63, e202317969). Thus, the synergy between In-TCPP sites for CO₂ reduction and In-O sites for H₂O oxidation enhances the overall photosynthetic efficiency.

In response to the reviewer's comments, we have included this discussion on Page 10, line 286 of the manuscript. "In-TCPP alone demonstrates a high selectivity for HCOOH production (81%), close to that of the heterostructure and higher than M68N (65%), underscoring the In-TCPP nanosheet shell's beneficial role in CO₂ adsorption and reduction. Meanwhile, the hydrophilic M68N provides effective sites for H₂O adsorption and oxidation to H₂O₂. **Additional experiments using sacrificial donors (Supplementary Fig. 29) confirm that H₂O as the electron donor achieves higher selectivity toward HCOOH, thus illustrating the advantages of the M68N@In-TCPP heterostructure in the overall CO₂ photoreduction reaction.** The origin of the carbon-containing products was further confirmed by ¹³C isotopic labelling and analysed using GC-MS and ¹³C NMR."

Comment 4. The defective In metal sites were formed in the MOF heterostructures to function as active sites for photocatalytic reaction. What about the stability of the defective sites? The EPR spectrum of the catalysts should be monitored to present the changes in oxygen vacancies during long-term test.

Response: We appreciate the reviewer's suggestion. To address this, we monitored the EPR spectra of M68N@In-TCPP heterostructures during prolonged photocatalytic tests (2 h, 5 h, and 20 h). The defective In sites, observed as a peak at $g = 2.0035$, showed a slight increase in intensity within the first 5 hours and then remained stable up to 20 hours (Supplementary Fig. 44). Additionally, the principal g -values of these sites did not change, confirming the stable structure and valence state of In in In₂O₄(OH)₂. These findings illustrate that the defective In sites remain stable during extended operation, ensuring sustained photocatalytic performance.

Supplementary Fig. 44. (a) Photocatalytic performance of M68N@In-TCPP after reaction 2, 5, and 20 h. (b) The evolution of O_v within the M68N@In-TCPP heterostructures during the reaction

As requested, we have added these results on Page 14 Line 412: “Upon exposure to light, the M68N@In-TCPP exhibited an increased peak intensity at $g = 2.0035$ (Fig. 3e), indicating light-driven electron transfer from the porphyrin unit. During the photocatalytic reaction, the defective In sites showed a slight increase in intensity within the first 5 hours and then remained stable up to 20 hours (Supplementary Fig. 44). Additionally, the principal g -values of these sites did not change, confirming the stable structure and valence state of In in $In_2O_4(OH)_2$. These findings illustrate that the defective In sites remain stable during extended operation, ensuring sustained photocatalytic performance. When we replaced N_2 with CO_2 , the signal intensity diminished, reflecting electron consumption by adsorbed CO_2 molecules.”

Comment 5. Although the experiments are thoroughly conducted, the structural stability of the MOF heterostructure needs further study, for example, the effect of H_2O_2 and $\bullet OH$ products on the decomposition of the organic linkers, the leaching of metal sites.

Response: We appreciate the insightful comment of the reviewer. To address the reviewer’s suggestion, we evaluated the photostability and structural integrity of the M68N@In-TCPP heterostructure over prolonged reaction times (Supplementary Fig. 36) and multiple cycles (Fig. 2g). PXRD patterns of fresh and reused samples (Supplementary Fig. 32) confirm that both MOFs retain their crystal structures after extended operation. SEM images (Supplementary Fig. 33) show that the core@shell morphology remains intact, and consistent FT-IR and XPS spectra (Supplementary Figs. 34 and 35) further support the chemical stability.

To confirm that no components leach or degrade, we analysed the reaction solution using 1H -NMR and ICP-MS. No characteristic signals of TCPP or NH_2 -BDC fragments were detected (Supplementary Fig. 36), and metal node loss was negligible, remaining below 0.19 wt% even after long-term reactions (Supplementary Table 6).

Supplementary Fig. 36. (a) The $^1\text{H-NMR}$ spectra of solution after 5, 10, and 20 h of reaction. The solution was collected by separating the M68N@In-TCPP for analysis. (b) The corresponding In content in the solution.

Supplementary Table 6. ICP-MS analysis of the In content in the solution.

Amount of indium in the eluent	$^a\text{M68N@In-TCPP}$
Before test	0
After 5 h of reaction	^b n. d.
After 10 h of reaction	1.84 μg (0.13 wt%)
After 20 h of reaction	2.73 μg (0.19 wt %)

Note: ^athe content of In in M68N@In-TCPP is 27.55 wt%. ^bnot detect.

These findings, now included on Page 11 line 311, demonstrate the robust frameworks of In-based MOFs and the good durability of M68N@In-TCPP during CO_2 conversion. “The intact crystal structure of each MOF in the heterostructure, confirmed by comparing PXRD patterns of fresh and used samples (Supplementary Fig. 32), and the uniform core@shell morphology of the recycled M68N@In-TCPP (Supplementary Fig. 33), along with consistent FT-IR and XPS spectra (Supplementary Figs. 34 and 35), further confirm the chemical stability. The reaction solution was detected by inductively coupled plasma mass spectrometry (ICP-MS) and $^1\text{H-NMR}$. There are no characterized signals of TCPP or $\text{NH}_2\text{-BDC}$ fragments after reactions and only less than 0.19 wt % of the metal nodes are leaching after long-term reaction (Supplementary Fig. 36 and Table 6). The robustness of In-based MOFs, derived from strong In-carboxyl coordination bonds and coherent MOF–MOF interfaces, ensures efficient and stable overall photosynthesis^{15,47}.”

Reviewer #2

Comment: In this manuscript, the authors report the synthesis of In-based MOF (In-porphyrin (In-

TCPP) nanosheets enveloping an In-NH₂-MIL-68 core) for simultaneous photocatalytic CO₂ reduction and H₂O oxidation. They found that the In-TCPP shell enhanced CO₂ adsorption and its subsequent photocatalytic reduction to produce HCOOH, while the In-O sites in the M68N core promoted H₂O oxidation to generate H₂O₂. They attributed the better performance of this heterostructure to the enhanced charge separation and transfer due to the formation of a Z-scheme band structure. However, the significance of this manuscript is minimized by the unclear selectivity and confusable processes of CO₂ reduction and water oxidation. Thus, this paper is not recommended for publication. Some specific comments are as follows.

Response: Thank you for acknowledging the concept presented in our manuscript. Achieving efficient photocatalytic CO₂-to-HCOOH reduction coupled with the direct formation of H₂O₂ is a significant challenge. In this study, we designed a core-shell M68N@In-TCPP heterostructure that enhances CO₂ adsorption and selective HCOOH production through the In-TCPP shell. Under sunlight and without any photosensitizer or sacrificial agent, the photocatalytic conversion of CO₂ to HCOOH on the In-TCPP shell reached 121.1 μmol g⁻¹ h⁻¹ with a high selectivity of 85.4%. Meanwhile, the high hydrophilicity of M68N facilitates H₂O adsorption and contributes to efficient H₂O oxidation to H₂O₂ with a production rate of 119.3 μmol g⁻¹ h⁻¹ and selectivity of 94.8%. The well-aligned HOMO level of M68N supports the oxidation reaction. Upon irradiation, photogenerated electrons and holes are effectively separated across the coherent interface between M68N and In-TCPP, directing electrons toward In-N sites for CO₂ reduction and holes toward In-O sites for H₂O oxidation. Thus, reduction and oxidation processes occur simultaneously on their respective sites.

In-situ DRIFTS analysis and DFT calculations further clarify the coupled mechanism of CO₂ reduction and H₂O oxidation. CO₂-to-HCOOH on the In-N sites proceeds through two hydrogenation steps: from *CO₂ to HCOO* and from HCOO* to HCOOH*. In parallel, H₂O oxidation at the In-O sites involves two successive steps, converting H₂O into •OH radicals. Once HCOOH is released, these two •OH radicals combine to form H₂O₂, thereby maintaining overall charge balance. This detailed mechanistic insight helps clarify the selectivity and the underlying processes, strengthening the significance and impact of our work.

Comment 1. One major concerning point is the selectivity of photocatalytic reduction and oxidation. As we known, H₂O₂ is easier to be reduced than CO₂, while HCOOH can also be oxidized by photogenerated holes. The authors should clearly explain why In-based MOF selectively reduce CO₂ instead of H₂O and H₂O₂, and selectively oxidize H₂O other than HCOOH and H₂O₂ in more detail.

Response: We appreciate the reviewer's comment, which helps us clarify the reaction selectivity in our system. Our findings show that the M68N@In-TCPP heterostructure selectively reduces CO₂ rather than H₂O or H₂O₂, and oxidizes H₂O rather than HCOOH or H₂O₂. This selectivity is closely linked to differences in adsorption behaviour and charge transfer pathways. The additional

content reinforces the inherent selectivity of M68N@In-TCPP and addresses the reviewer's concerns.

First, the enhanced CO₂ adsorption on In-TCPP sites drives selective CO₂ reduction. Even with a small (10%) molar ratio of In-TCPP, the CO₂ uptake capacity doubles from 10.2 to 21.8 cm³ g⁻¹ at 298 K and 1.0 bar, while the water contact angle increase from 26.6 to 69.8°. This improved CO₂ uptake arises from In-TCPP's π -conjugated porphyrin rings and polar pyrrole groups with strong CO₂ affinity. Under illumination, electrons move from M68N's LUMO to In-TCPP's HOMO, where they recombine with holes, ensuring efficient charge separation. As a result, the In-TCPP component becomes the active reduction site, favouring CO₂ reduction over H₂O or H₂O₂ due to CO₂'s stronger adsorption and higher local concentration.

In contrast, the M68N component adsorbs H₂O strongly due to its hydrophilicity, making it accessible to photogenerated holes for oxidation. HCOOH, formed at In-TCPP sites and low concentrations relative to H₂O, is not oxidized. Thus, the system exploits the different adsorption affinities and localised electronic structures of M68N and In-TCPP to achieve selective CO₂ reduction and H₂O oxidation, while suppressing H₂O₂ reduction and HCOOH oxidation.

To further support these conclusions, we conducted comparative experiments (Supplementary Figs. 22–25 and Table 5). The linear sweep voltammetry measurements in CO₂ or Ar indicate a strong preference for CO₂ reduction over the hydrogen evolution reaction (Supplementary Fig. 22). In Ar, only small amounts of H₂ and H₂O₂ were detected, confirming that M68N@In-TCPP barely favours H₂O reduction (Supplementary Fig. 23). In-based materials commonly favour *COOH intermediates, enabling selective formate (HCOOH) production during CO₂ reduction (*J. Am. Chem. Soc.* 2021, 143, 6877; *Chem. Soc. Rev.* 2020, 49, 6884). As reported, the difference in required potentials for CO₂ reduction and H₂ evolution ($U_L(\text{CO}_2) - U_L(\text{H}_2)$) correlates with observed selectivity based on the density functional theory (DFT) studies (*J. Am. Chem. Soc.* 2017, 139, 8329; *Angew. Chem. Int. Ed.* 2020, 59, 22465). A positive $U_L(\text{CO}_2) - U_L(\text{H}_2)$ for In-based catalysts suggests higher CO₂ reduction selectivity over HER. To verify the CO₂ reduction selectivity, we replaced CO₂ by Ar to compare the activity of CO₂ and H₂O reduction.

Supplementary Fig. 22. The Linear sweep voltammetry (LSV) curve of M68N@In-TCPP in CO₂ and Ar atmosphere with KOH (0.1 M) solution.

Supplementary Fig. 23. The HCOOH, CO, H₂, and H₂O₂ production rates of the control experiments under different atmospheres using M68N@In-TCPP as catalyst.

Additional tests show that H₂O₂ is neither effectively oxidized to O₂ nor significantly reduced to H₂O in Ar after 20 h of light irradiation (Supplementary Fig. 24). Replacing H₂O with CH₃CN and adding HCOOH leads to minimal HCOOH oxidation, again at a much lower rate than CO₂ reduction (Supplementary Fig. 25 and Table 5). These results confirm the preferential reduction of CO₂ to HCOOH over H₂O₂ or H₂O reduction, and the preferential oxidation of H₂O over HCOOH or H₂O₂ oxidation.

Supplementary Fig. 24. Changes in H₂O₂ concentration after introducing H₂O₂ (1.2 mM) into the photocatalytic system with M68N@In-TCPP in Ar. H₂O₂ was dissolved in CH₃CN to avoid the interference by H₂O. (a) H₂O₂ concentration during light irradiation, (b) the corresponding UV adsorption spectra of I₃⁻ in the presence of H₂O₂.

Supplementary Fig. 25. Changes in HCOOH concentration after introducing HCOOH (1.3 mM) into the photocatalytic system with M68N@In-TCPP in Ar. HCOOH was dissolved in CH₃CN to avoid the interference by H₂O. (a) HCOOH concentration during light irradiation, (b) the corresponding ¹H-NMR signal of the solution.

Supplementary Table 5. Comparative experiments under different conditions for the artificial photosynthetic overall reaction of M68N@In-TCPP.

Reactant	Atmosphere	20 h of irradiation	Amount of the product after 20 h of reaction	
			Reduction product	Oxidation product
H ₂ O	CO ₂	-	HCOOH (12.5 μmol)	H ₂ O ₂ (12 μmol)
			CO (2.24 μmol)	O ₂ (0.54 μmol)
H ₂ O	Ar	-	H ₂ (1.98 μmol)	H ₂ O ₂ (2.45 μmol)
			^a H ₂ O ₂ (1.2 mM)	^c O ₂ (n.d.)
in CH ₃ CN	Ar	H ₂ O ₂ (1.16 mM)		H ₂ (n.d.)
^b HCOOH (1.3 mM)	Ar	HCOOH (1.16 mM)	CO ₂ (n.d.)	CO (0.53 μmol)
				H ₂ (n.d.)

Note: ^aadding H₂O₂ with the concentration equal to the H₂O oxidation yields over M68N@In-TCPP. ^badding HCOOH with the concentration equal to the CO₂ reduction yields over M68N@In-TCPP. ^cn.d.: not detected.

In response to the reviewer's suggestion, we have revised the manuscript (Page 8, line 228) to include these explanations. "Importantly, the system did not show detectable H₂ evolution, indicating a preferential reaction of CO₂ reduction over H₂O, attributed to the higher overpotential required for H₂ evolution reaction (HER) compared to CO₂ reduction over In-based catalysts. This result can be verified by the linear sweep voltammetry curves of M68N@In-TCPP in Supplementary Fig. 22. Strong adsorption of *COOH intermediates at In sites also helps suppress the competing HER, corroborated by the previously reported low HER activity work^{30,31}. In the absence of CO₂, only small amounts of H₂ and H₂O₂ were detected in argon(Ar) atmosphere, confirming the unfavorable H₂O reduction process on M68N@In-TCPP (Supplementary Fig. 23). The control experiments by the additional introduction of H₂O₂ or HCOOH indicate the preferential reduction of CO₂ to HCOOH over H₂O₂ or H₂O reduction, and the preferential oxidation of H₂O over HCOOH or H₂O₂ oxidation (Supplementary Figs. 24, 25, and Table 5)^{30,31}."

Comment 2. The generation of HCOOH and H₂O₂ should be competitive with water as the only proton source. Although the authors measured the production of HCOOH and H₂O₂ in this core-shell M68N@In-TCPP heterostructure photocatalysis, their quantum yields were very low. Thus, the authors are suggested to further understand the reaction processes.

Response: We appreciate the reviewer's suggestion to clarify the reaction processes. In our system, the half-reactions are as follows:

Here, the protons (H⁺) generated from water oxidation are consumed by the CO₂ reduction, ensuring that the formation of HCOOH and H₂O₂ is not directly competitive. Since water is abundant, it is not depleted by these reactions.

The overall artificial photosynthesis for HCOOH and H₂O₂ production from CO₂ and H₂O is a promising approach for sunlight-driven CO₂ recycling. However, only a few representative examples reported the overall photosynthesis because the efficiency of photocatalysts had been hampered by the chemical inertness of CO₂ and the sluggish kinetics of water oxidation, limiting the efficacy of many existing systems to one half-reactions often reliant on additional photosensitizers and sacrificial agents (Table 1 and Supplementary Table 1). By constructing the coherent interfaces of the core@shell MOFs, the In-TCPP shell in MOFs heterostructure improves CO₂ adsorption capabilities and visible light absorption to enhance the photocatalytic CO₂ reduction, while the In-O sites in M68N core efficiently catalyse H₂O oxidation. Thus the apparent quantum yield (AQY) of M68N@In-TCPP at 420 nm is 0.16%, placing it the high level among the representative MOF/COF-based catalysts for overall photosynthesis in liquid-solid systems. Furthermore, the AQY remains as high as 0.10% in the 500–600 nm range. These results demonstrate the potential of the M68N@In-TCPP catalyst to harness solar energy effectively for environmental remediation and sustainable fuel production.

Comment 3. Abstract: The argument that “the inherent catalytic inefficiency of single-structure In-based MOFs” is too general, and the solution of “constructing a heterostructure comprising two MOF structures” for this question is quite rough and simple.

Response: We thank the reviewer for this valuable feedback and have carefully checked the above-mentioned statement in the introduction part. Now the more detailed descriptions have been presented on Page 4 line 85 as follows: “However, single-structure In-based MOFs often suffer from limited catalytic activity of overall photosynthesis, primarily due to the quenching of photogenerated charges and difficulty in integrating distinct active sites necessary for both half-reactions. To address these challenges, we constructed a heterostructure comprising two MOF structures with light-harvesting units for broad visible light response and distinct metal catalytic sites for reduction and oxidation reactions. The difference in growth kinetics of indium (In) with

two types of organic linkers enables a one-pot approach that reduces the synthesis complexity and facilitates the formation of a coherent interface between the two MOF lattices by sharing the same metal atoms. This design leverages unique organic linkers to establish built-in electric fields and atomic-scale charge-transfer pathways, thereby promoting directed charge separation and enhancing overall catalytic performance under visible-light irradiation without relying on external photosensitizers, noble metals, or sacrificial agents (Table 1 and Supplementary Table 1).”

Comment 4. As both the CO₂ reduction and H₂O₂ production require protons, the authors should consider the effect of pH on the performance of M68N@In-TCPP.

Response: We appreciate the reviewer’s suggestion and have studied the effect of pH on the performance of M68N@In-TCPP. The photocatalytic CO₂ reduction and H₂O oxidation activity is dependent on the pH value in a wide range (pH 3–12). The pH-dependent experiment reveals that M68N@In-TCPP exhibits optimal HCOOH and H₂O₂ yields at pH = 7 (Supplementary Fig. 28). The production rates of HCOOH and H₂O₂ remarkably decreased in the reaction system with a high concentration of H⁺ or OH⁻ (pH = 3 or 12). The CO₂ reduction involves 2 protons and 2 electrons transfer process. The HCOOH formation is limited by the poor CO₂ solubility in water at lower pH and almost suppressed by the inefficient proton supply at higher pH (*Energy Environ. Sci.* 2016, 9, 2177; *J. CO₂ Util.* 2023, 70, 102428). It is noted that the net consumptions of H⁺ and OH⁻ in the overall reactions are zero. The H₂O₂ production involving H₂O as proton donor is also slowed by the high concentration of H⁺ or OH⁻ (*J. Am. Chem. Soc.* 2024, 146, 21147). The similar pH-dependent trend in each half-reaction suggests the simultaneous CO₂ reduction and water oxidation on M68N@In-TCPP photocatalysts.

Supplementary Fig. 28. The photocatalytic CO₂ reduction performance of M68N@In-TCPP in water at different pH values.

In response to the reviewer’s suggestion, we have added the corresponding description on Page 10 line 277: “Regarding CO₂ reduction, In-TCPP displayed high selectivity for HCOOH

production (81%), closely approaching that of the heterostructure and higher than M68N (65%), highlighting the beneficial role of the In-TCPP nanosheet shell in CO₂ adsorption and reduction, while the hydrophilic M68N provides effective sites for H₂O adsorption and oxidation to H₂O₂. The pH-dependent experiment reveals that M68N@In-TCPP exhibits optimal HCOOH and H₂O₂ yields at pH = 7 (Supplementary Fig. 28). The production rates of HCOOH and H₂O₂ remarkably decreased in the reaction system with a high concentration of H⁺ or OH⁻ (pH = 3 or 12). The CO₂ reduction involves 2 protons and 2 electrons transfer process. The HCOOH formation is limited by the poor CO₂ solubility in water at lower pH and almost suppressed by the inefficient proton supply at higher pH. It is noted that the net consumptions of H⁺ and OH⁻ in the overall reactions are zero. The H₂O₂ production involving H₂O as proton donor is also slowed by the high concentration of H⁺ or OH⁻.⁴⁵ The similar pH-dependent trend in each half-reaction suggests the simultaneous CO₂ reduction and water oxidation on M68N@In-TCPP photocatalysts.”

Comment 5. The stability of the material should be evaluated by characterizing the used photocatalysts, as the organic frameworks can serve as the proton donors.

Response: We appreciate the reviewer’s suggestion. Our characterizations of the used M68N@In-TCPP catalyst confirm its stability. As shown in Supplementary Figs. 32–35, the crystal structure and morphology remain intact after use. There are no characterized signals of TCPP or NH₂-BDC fragments after reactions and only less than 0.19 wt % of the metal nodes are leaching after long-term reaction (Supplementary Fig. 36 and Table 6). To further rule out the possibility that the MOF’s organic linkers serve as proton donors, we conducted control experiments by replacing H₂O with CH₃CN. Under these conditions, no HCOOH was detected, and ¹H-NMR showed no signals attributable to the linker, confirming that the organic framework does not contribute protons (Supplementary Fig. 37). These findings indicate that the MOF framework remains stable, with no detectable release of linker fragments, and that H₂O is the sole proton source.

The data, including PXRD patterns, SEM images, FT-IR and XPS spectra (Supplementary Figs. 32–35), and ICP-MS and ¹H-NMR analyses, confirm the catalyst’s chemical and structural stability. The robustness of In-based MOFs, derived from strong In-carboxyl coordination bonds and coherent MOF–MOF interfaces, ensures efficient and stable overall photosynthesis.

Supplementary Fig. 36. (a) $^1\text{H-NMR}$ spectra of solution after 5, 10, and 20 h of reaction. The solution was collected by separating the M68N@In-TCPP for analysis. (b) The corresponding In content in the solution.

Supplementary Fig. 37. (a) Photocatalytic performance of M68N@In-TCPP in CH_3CN . (b) $^1\text{H-NMR}$ spectra of solution after 20 h of reaction.

Supplementary Table 6. ICP-MS analysis of the In content in the solution.

Amount of indium in the eluent	$^a\text{M68N@In-TCPP}$
Before test	0
After 5 h of reaction	$^b\text{n. d.}$
After 10 h of reaction	1.84 μg (0.13 wt%)
After 20 h of reaction	2.73 μg (0.19 wt%)

Note: a the content of In in M68N@In-TCPP is 27.55 wt%. b not detect.

We have added a corresponding description on Page 11 line 311. “The intact crystal structure of each MOF in the heterostructure, confirmed by comparing PXRD patterns of fresh and used samples (Supplementary Fig. 32), and the uniform core@shell morphology of the recycled M68N@In-TCPP (Supplementary Fig. 33), along with consistent FT-IR, XPS spectra (Supplementary Fig. 34 and 35), further confirm the chemical stability. The reaction solution was detected by inductively coupled plasma mass spectrometry (ICP-MS) and $^1\text{H-NMR}$. There are no characterized signals of TCPP or $\text{NH}_2\text{-BDC}$ fragments after reactions and only less than 0.19 wt % of the metal nodes are leaching after long-term reaction (Supplementary Fig. 36 and Table 6). To further rule out the possibility that the MOF’s organic linkers serve as proton donors, we conducted control experiments by replacing H_2O with CH_3CN . Under these conditions, no detectable HCOOH and no signals attributable to the linker confirm that the organic framework does not contribute protons (Supplementary Fig. 37). These findings indicate that the MOF framework remains stable, with no detectable release of linker fragments, and that H_2O is the sole proton source. The robustness of In-based MOFs, derived from strong In-carboxyl coordination bonds and coherent MOF–MOF interfaces, ensures efficient and stable overall

photosynthesis^{15,47}.”

Comment 6. Normally a Z-scheme photocatalytic system should have electron shuttles, which are absent in the M68N@In-TCPP. This catalyst is a typical heterojunction; but does not meet the characteristics of Z-scheme photocatalyst.

Response: Thank you for the comment. We acknowledge that multiple types of Z-scheme systems exist and our catalyst aligns more closely with the characteristics of a “direct Z-scheme” architecture according to the literatures. Generally, a direct Z-scheme heterojunction is built at the interface of two photocatalysts with matched energy levels. The interfacial contact between them will cause the down-bending of one photocatalyst with positive band energy and upward bending of another photocatalyst with negative band energy, leading to the formation of interfacial electric field (IEF) for improving charge separation and transfer efficiency (Fig. R1, *Adv. Mater.* 2022, 34, 2106807; *Angew. Chem. Int. Ed.* 2024, 63, e202308597; *Adv. Funct. Mater.* 2024, 2416556).

Figure R1. Schematic illustration of charge transfer mechanism in the representative Z-scheme photocatalysts via the internal electric field (IEF). (a) Schematic diagram of Z-scheme Zn-TCPP/THPP photocatalyst; (b) Z-scheme MIL-125(Ti)-50@TiCe-MOF, (c) Schematic diagram of interface charge transfer route for NDINH/PDINH Z-scheme heterostructure. (*Adv. Mater.* 2022, 34, 2106807; *Adv. Funct. Mater.* 2024, 2416556; *Angew. Chem. Int. Ed.* 2024, 63, e202308597)

Since there are multiple types of Z-scheme and other analogous names of heterojunction system, we have deleted the specific name as “Z-scheme” and explained the formation of M68N@In-TCPP heterojunction photocatalysts in the revised manuscript.

Comment 7. The strong absorption peak at 1266 cm⁻¹ was attributed to the O–H deformation vibration (•OH). Was this •OH a radical or adsorbed OH group? It is very difficult for 1e⁻ WOR to form •OH radical.

Response: We appreciate the reviewer’s suggestion to clarify the water oxidation reaction processes. The O–H deformation vibration at 1320 cm⁻¹ is from the adsorbed H₂O₂ on catalyst surface (*Nat. Commun.* 2022, 13, 4592; *Appl. Spectrosc.* 2003, 57, 574.). In general, the water

oxidation reaction (WOR) involves a $2e^-$ process for H_2O_2 production and a $4e^-$ process for O_2 production. The O_2 production exhibits sluggish kinetics, compared to the H_2O_2 production. The H_2O -to- H_2O_2 oxidation is likely a two-step $2e^-$ WOR process with $\bullet OH$ radical as intermediates, i.e., $H_2O + h^+ = \bullet OH + H^+$; $\bullet OH + \bullet OH = H_2O_2$. In general, electron paramagnetic resonance (EPR) with 5,5-dimethyl-1-pyrroline N-oxide (DMPO) is used to study the in situ generated $\bullet OH$ radicals during H_2O_2 photosynthesis (*Nat. Synth.* 2024, doi: 10.1038/s44160-024-00644-z; *J. Am. Chem. Soc.* 2023, 145, 50, 27757; *Angew. Chem. Int. Ed.* 2022, 61, e202200413; *Adv. Energy Mater.* 2022, 12, 2201466). Therefore, we supplemented the EPR experiments to study the possible intermediates for H_2O_2 formation in this work. In the absence of photocatalyst, there were no peaks corresponding to $\bullet OH$ in the EPR spectrum. During the photocatalytic reaction on M68N@In-TCPP, a distinct 1:2:2:1 quadruplet peak of DMPO- $\bullet OH$ adducts in EPR spectrum suggests the $\bullet OH$ as the intermediate species for H_2O_2 formation (Supplementary Fig. 46).

Supplementary Fig. 46. EPR spectra of DMPO- $\bullet OH$ with or without M68N@In-TCPP photocatalyst under light irradiation.

In response to the reviewer's suggestion, we have added the related explanations on Page 15 line 439 of the manuscript: "During the photocatalytic process, a strong absorption peak at 1320 cm^{-1} arises from the O-H deformation vibration of the adsorbed H_2O_2 product¹⁶. EPR with 5,5-dimethyl-1-pyrroline N-oxide (DMPO) is used to study the possible intermediates for H_2O_2 formation. In the absence of photocatalyst, there were no peaks corresponding to $\bullet OH$ in the EPR spectrum. During the photocatalytic reaction on M68N@In-TCPP, a distinct 1:2:2:1 quadruplet peak of DMPO- $\bullet OH$ adducts in EPR spectrum suggests the $\bullet OH$ as the intermediate species for H_2O_2 formation (Supplementary Fig. 46)."

Reviewer #3

Comment: This work by Cai et al developed a composite photocatalyst with two MOF

nanomaterials based on indium. The formation process of the titled heterostructure is interesting with good elucidation. Moreover, the high affinity between two MOF nanostructures enables synergistic effects in achieving artificial photosynthesis with good yields in CO and H₂O₂. The mechanistic studies also verify the photocatalytic pathways to some extent with rational experimental/computational design. Overall, I recommend the publication of this work on Nature Communications after addressing the following issues.

Response: We sincerely thank the reviewer for recognizing the significance of our work and for the constructive feedback. We appreciate the reviewer's insightful comments and have incorporated their suggestions to strengthen the manuscript. Below, we provide a point-by-point response to each of the reviewer's comments.

Comment 1. I suggest putting the less important EDX mapping of Figure 1 into SI for higher clarity in presentation.

Response: Thank the reviewer for the suggestion. We have relocated Fig. 1d and 1e to the Supporting Information, where they are now labeled as Supplementary Figs. 5.

Supplementary Fig. 5. SEM images of M68N@In-TCPP and the corresponding element mapping distributions.

Comment 2. In Figure 1m, the XPS of In seems to be very smooth. Did the authors give the smoothed data? Original data points should nonetheless be provided like the XPS O 1s results.

Response: Thank you for your insightful question. We did not apply any smoothing to the XPS data presented in Figure 1m. The smooth appearance of the In 3d spectrum is due to the high In content as 27.55 wt% in M68N@In-TCPP producing the strong signal intensity. Meanwhile, the repeatedly measurement in the related region provides sufficient data points for an accurate representation.

Comment 3. The reported record-high activity in CO and H₂O₂ yields compared to other literature

is not very telling since the light intensity of the reported system far exceeds one sun (nearly 10 and 40 times). The utilized light intensity and apparent quantum yields should be supplied in Table 1 with abundant discussion on this issue.

Response: Thank the reviewer for highlighting this important issue and for the reviewer's valuable suggestion. We apologize for the incorrect description of the light intensity in our original manuscript. Specifically, the reported values of 0.94 W and 4.19 W refer to the input light power, while the correct light intensities are 0.3 W cm⁻² and 1.3 W cm⁻², respectively. We have updated Table 1 to include the accurate light intensity and apparent quantum yield (AQY) values.

We agree that the reported light intensities significantly exceed one sun. In fact, these intensities are achievable using a simple solar parabolic concentrator. Importantly, even under these higher light intensities, the photocatalytic performance of the M68N@In-TCPP heterostructure remains among the top catalysts for overall photosynthesis in liquid-solid systems.

Now we have revised the manuscript to provide a comprehensive discussion on this matter. The corresponding description on Page 12 line 334 has been updated as follows: "Consequently, the apparent quantum yield (AQY) of the M68N@In-TCPP heterostructure at 420 nm was measured to be 0.16% with light intensity of 9.6 mW cm⁻² (Fig. 2i, Supplementary Table 4), significantly exceeding the individual contributions of M68N (0.019%) and In-TCPP (0.014%). The AQY of the physical mixture of M68N and In-TCPP (M68N+In-TCPP) at 420 nm is 0.015%, only a tenth of the core@shell M68N@In-TCPP heterostructure. This highlights the heterostructure's superior photocatalytic performance and demonstrates the practical feasibility of using sustainable sunlight with light intensities ranging from 0.3 to 1.3 W cm⁻², surpassing those used in most recently reported CO₂ photoreduction systems (Table 1 and Supplementary Table 1)."

Table 1. Performance comparison of known MOF/COF-based catalysts for overall photosynthesis and half CO₂ reduction reaction.

Photocatalyst	Experiment conditions	Reduction product (μmol g ⁻¹ h ⁻¹)	Oxidation product (μmol g ⁻¹ h ⁻¹)	Add SA and/or PS ^a	AQY (%)	Ref.
Fe-In-TCP (Porphyrin-based MOF)	300W Xe lamp (>400 nm) CO ₂ , H ₂ O	HCOOH (17.6)	H ₂ O ₂ (13.04)	-	n.r.	20
MCOF-Ti ₆ Cu ₃	300W Xe lamp CO ₂ , H ₂ O (>400 nm, 0.4 W cm ⁻²)	HCOOH (169.0)	O ₂ (n.r. ^b)	-	n.r.	21
NNU-31-Zn (Zn-based COF ^c)	300W Xe lamp (>400 nm, 0.4 W cm ⁻²) CO ₂ , H ₂ O	HCOOH (26.3)	O ₂ (12.6)	-	0.035 (420 nm, 29 mW cm ⁻²)	22
Bi-TTCOF-Zn	300W Xe lamp (>420 nm, 0.4 W cm ⁻²) CO ₂ , H ₂ O	CO (11.6)	O ₂ (5.8)	-	n.r.	23

PCN-601 (Ni-based Porphyrin MOF)	300W Xe lamp (>410 nm, 0.25 W cm ⁻²) CO ₂ , H ₂ O	CO (6.0) CH ₄ (10.1)	H ₂ O ₂ (37.5)	-	0.064 (405 nm, 64 mW cm ⁻²)	24
Eu-bpy-Ru-CuCl ₂ ^d	300W Xe lamp (>420 nm, 0.15 W cm ⁻²) CO ₂ , MeCN/H ₂ O	n.d ^e	n.d	HCOOH (304)	n.r.	25
COF-367-Co ^{III} (Co-based Porphyrin COF)	300W Xe lamp (>380 nm), CO ₂ , MeCN/H ₂ O	n.r	n.r	HCOOH (93)	n.r.	26
UiO67-Ir-Cou 6/Cu ^f	300W Xe lamp (>420 nm, 0.2 W cm ⁻²) CO ₂ , MeCN/H ₂ O	n.d	n.d	HCOO ⁻ (408)	n.r.	27
Ru(phen) ₃ -Eu-MOF	300W Xe lamp (>420 nm) CO ₂ , MeCN/H ₂ O	n.d.	n.d	HCOO ⁻ (960)	n.r.	28
MOF-808-EDTA	300W Xe lamp (>420 nm, 1.58 W cm ⁻²) CO ₂ , MeCN/H ₂ O	n.d	n.d.	HCOOH (167)	n.r.	29
M68N@In-TCPP	300W Xe lamp (>400 nm, 0.3 W cm ⁻²) Sunlight ^g (1.3 W cm ⁻²) CO ₂ , H ₂ O	HCOOH (121.1) CO (22.3) HCOOH (397.5) CO (61.2)	H ₂ O ₂ (119.3) O ₂ (5.9) H ₂ O ₂ (321.2) O ₂ (not detected)	-	0.16 (420 nm, 9.6 mW cm ⁻²)	This work

^aSA: sacrificial agents; PS: photosensitizers; ^bn.r.: not reported; ^cCOF: covalent organic framework; ^dEu-bpydc, bpydc = 2,2'-bipyridine-5,5'-dicarboxylate, integrate with Ru(bpy)₃ photosensitizer (PS); ^en.d.: not detect; ^fUiO67-Ir-Cou 6: UiO-67 MOF integrate with Ir-ppy and coumarin 6 PS; ^gSunlight: The sunlight intensity was gathered and enhanced by the condenser.

Comment 4. Why CO and O₂ product yields are absent in Figure 2a and related discussion, while the two products could be detected in another system as shown in Figures 2a and 2g? This is also not consistent with Table 1 where no oxygen yield was given.

Response: Thank the reviewer for highlighting this important issue. We have addressed the concerns as follows:

We have now included the CO product yield data in Figure 2a to provide a complete overview of the reaction products.

The O₂ yield in Figure 2b has been produced by irradiating the photocatalytic system in an indoor online photocatalytic test setup and automatically injected into a gas chromatography system (Agilent 8860, USA) (Supplementary Fig. 16). The combination of indoor online setup and GC can avoid the deviation by O₂ in external atmosphere. The results in Figure 2a were obtained by the outdoor photocatalytic equipment under natural sunlight. The O₂ yields were not initially

reported in Figure 2a and Table 1 due to the difficulty in accurately detecting trace amounts of O₂ in the sunlight-driven CO₂ photoreduction system using offline injection methods. The low concentrations of O₂ made reliable quantification challenging under these experimental conditions. To maintain consistency, we have explained the absent O₂ result under sunlight condition as not detected in Table 1.

Fig. 2a. High performance of sunlight-driven photoreduction CO₂ and coupling H₂O oxidation. (a) Practical performance test and corresponding STC conversion efficiency of M68N@In-TCPP photocatalyst under 4 h of natural sunlight irradiation between January 10 and 13th, 2024.

We have revised the manuscript to clarify these points, ensuring that the presence and detection challenges of O₂ are transparently discussed. Page 8 line 202 “The H₂O₂ yield was slightly lower at 321.2 μmol g⁻¹ h⁻¹ compared to the HCOOH yield, potentially due to its decomposition under intense light conditions. The O₂ yields were not reported due to the difficulty in accurately detecting trace amounts of O₂ in the sunlight-driven CO₂ photoreduction system using offline injection methods.”

This includes specifying the conditions under which O₂ was measured and the reasons for its absence in certain figures and tables. These revisions enhance the clarity and completeness of our reporting, addressing the inconsistencies and providing a more comprehensive understanding of the photocatalytic performance of M68N@In-TCPP. Thank the reviewer again for the constructive feedback.

Comment 5. The detection details of O₂ were not mentioned in Methods.

Response: Thank you for pointing out the omission of detection details for O₂ in the Methods section. To ensure accurate detection of gas products, including CO and O₂, samples were introduced via an online sampling port, minimizing interference from atmospheric O₂. The gas chromatography system (Agilent 8860, USA) was used for analysis, as illustrated in Supplementary Fig. 16.

We have updated the Methods section accordingly and included the O₂ detection details, along with the corresponding standard curves, in the Supplementary Information (Supplementary Fig. 16). Specifically, the revision on Page 22, line 669 states: “Photocatalytic gas products were

analyzed using a gas chromatography system (Agilent 8860, USA) equipped with an online test setup (Supplementary Fig. 16). The corresponding standard curve of O₂ is provided in Supplementary Fig. 17.” We believe these revisions address the reviewer's concern and enhance the clarity of the manuscript.

Supplementary Fig. 16. The system used for photocatalytic CO₂ reduction operates within a closed gas circulation setup. Inside the green box is a pressure gauge that monitors the system's pressure, which can be set to zero before introducing CO₂ gas (82.98 kPa). The red box contains the gas detection pathway for online sampling. The purple line represents the carrier gas (Ar), which mixes with the product gas from the red pathway before being injected into the GC for analysis.

Supplementary Fig. S17. The standard curve of detecting O₂ by the GC (Agilent-8860, USA)

Comment 6. In Figure 3f and 3g, what is the meaning of the 0.2 bars which are absent in Figure 3h and 3i?

Response: Thank you for your insightful comment. The 0.2 bar in Fig. 3f and 3g represents the intensity of the signal in the in-situ DRIFTS analysis. To improve clarity and ensure consistency, we have added the signal intensity values in Figure 3 of the revised manuscript. We believe this adjustment will make the data more comprehensible for readers.

Fig. 3. In situ DRIFTS spectra from 1300 to 1800 cm^{-1} . Reactants adsorption on (f) M68N and (g) M68N@InTCPP in the dark. Intermediates evolution on (h) M68N and (i) M68N@InTCPP during photocatalytic reaction.

Comment 7. If the decay was claimed as single-component, there will be no two τ values will be presented. Two components could be more suitable in discussing Supplementary Fig. 39.

Response: Thank you for bringing this to our attention. We agree that the description of the decay as single-component was incorrect. In the revised manuscript, we have updated the fitting to reflect two components and adjusted the discussion accordingly. We have carefully reviewed the entire manuscript, corrected all related errors, and highlighted the revisions for clarity.

Page 19 line 560: “Conversely, the Q-band emission presents a **two-component decay** profile with a lifetime of 2.55 ns ($\tau_1=14.85$, 0.01%; $\tau_2=2.55$, 99.99%), where the longer τ_1 indicates the photoinduced charge transport and the shorter τ_2 corresponds to porphyrin-localized exciton recombination (Supplementary Fig. 56)²⁴.”

Comment 8. I am curious why the produced H_2O_2 would not oxidize HCOOH, two of which are dissolved in water? A discussion needs to be given accordingly.

Response: Thank the reviewer for your valuable feedback. We understand your concern regarding the potential oxidation of HCOOH by the produced H_2O_2 in our system. To address this, we conducted additional experiments to demonstrate the stability of both H_2O_2 and HCOOH when

present together.

We added H₂O₂ (1.2 mM) and HCOOH (1.3 mM), with the concentration equal to the H₂O oxidation and CO₂ reduction yields, to the system without the catalyst and allowed the reaction to proceed for 20 hours under light irradiation. The results showed that only 1.5% of HCOOH decomposed to CO, and 1.2% of H₂O₂ was reduced to H₂O. This indicates that both H₂O₂ and HCOOH remain largely stable and coexist in the system.

Furthermore, the existing literatures support our findings. For instance:

- Lin et al. reported the coexistence of HCOOH and H₂O₂ during the photoreduction of CO₂ and H₂O, achieving production rates of 17.7 μmol·g⁻¹·h⁻¹ for HCOOH and 13.4 μmol·g⁻¹·h⁻¹ for H₂O₂ (*Angew. Chem. Int. Ed.* 2022, 62, e202111622).
- Ishihara et al. synthesized CH₃COOH, HCOOH (1.5 μmol·g⁻¹·h⁻¹), and H₂O₂ (35.2 μmol·g⁻¹·h⁻¹) via photocatalytic CO₂ reduction (*J. Am. Chem. Soc.* 2018, 140, 6474).
- Huang et al. achieved the photooxidation of CH₄ to HCOOH at a rate of 0.49 mmol·g⁻¹·h⁻¹ by using H₂O₂ as the main oxidant (*Nat. Commun.* 2022, 13, 6677).
- Wu et al. utilized 0.5 M H₂O₂ to convert CH₄ to HCOOH with a rate of 4.7 mmol·g⁻¹·h⁻¹ and over 90% selectivity (*Angew. Chem. Int. Ed.* 2021, 60, 8889).
- Tang et al. achieved direct CH₄ oxidation to HCOOH with a rate of 4.7 mmol·g⁻¹·h⁻¹ and over 97.9% selectivity by adding 30% H₂O₂ (*Angew. Chem. Int. Ed.* 2021, 60, 5811).
- Deng et al. converted CH₄ to HCOOH at a rate of 6.13 mmol·g⁻¹·h⁻¹ using H₂O₂ as the main oxidant (*J. Catal.*, 2024, 432, 115452).

These studies collectively demonstrate that H₂O₂ and HCOOH can coexist stably under light irradiation conditions.

In response to your suggestion, we have included a detailed discussion in the revised manuscript on Page 9 line 271: “This indicates that the H₂O oxidation is contingent upon electron consumption by CO₂ at In-TCPP’s reductive sites. By adding H₂O₂ and HCOOH aqueous solution, negligible decomposition of H₂O₂ and HCOOH was observed after the reaction (Supplementary Fig. 27), demonstrating that the liquid products H₂O₂ and HCOOH can remain stable and coexist in the system. Regarding CO₂ reduction, In-TCPP displayed high selectivity for HCOOH production (81%).”

We believe these additions adequately address your concern and enhance the clarity of our manuscript.

Supplementary Fig. 27. (a) Changes in HCOOOH and H₂O₂ concentration under light irradiation, (b) the corresponding ¹H-NMR spectra of the solution.

Comment 9. Recent examples on artificial photosynthesis, e.g., *Angew. Chem. Int. Ed.* 2024, 63, e202401344, *J. Am. Chem. Soc.* 2024, 146, 17773, etc., are suggested for citation.

Response: Thank you for your valuable suggestion. We have incorporated the recommended references into our manuscript to enhance the discussion on artificial photosynthesis. Specifically, we have added citations to the following sections on Page 2 line 54: “However, the inherent chemical stability of CO₂ and the slow oxidation of water necessitate the development of highly effective photocatalysts capable of driving these reactions efficiently^{9,10} (*Angew. Chem. Int. Ed.* 2024, 63, e202401344; *J. Am. Chem. Soc.* 2024, 146, 17773).” line 75: “Finally, for the industrial application of catalytic overall photosynthesis, it is crucial to harvest sufficient solar energy in the visible light range¹⁰ (*J. Am. Chem. Soc.* 2024, 146, 17773).”

These additions provide relevant context and support the significance of our research within the current advancements in the field of artificial photosynthesis.

Reviewer #4

Comment: I co-reviewed this manuscript with one of the reviewers who provided the listed reports. This is part of the Nature Communications initiative to facilitate training in peer review and to provide appropriate recognition for Early Career Researchers who co-review manuscripts.

Reviewer #5

Comment: An indium-based MOF heterostructure was synthesized using a one-pot method, which serves as an effective photocatalyst for simultaneous CO₂ reduction and water oxidation to yield

formic acid ($397.5 \mu\text{mol g}^{-1} \text{h}^{-1}$) and hydrogen peroxide ($321.2 \mu\text{mol g}^{-1} \text{h}^{-1}$), respectively. This manuscript describes a balanced work and is noteworthy. I am happy to recommend this paper for publication in Nature Communications. However, some following concerns need to be addressed:

Response: We sincerely thank the reviewer for their positive feedback and for recognizing the significance of our work. We are also grateful for the constructive comments and valuable suggestions, which have helped us enhance the quality of the manuscript. Below, we have provided detailed responses to each of the reviewer's comments.

Comment 1. The manuscript synthesized an interesting core-shell M68N@In-TCPP heterostructure through controlling the crystallization kinetics between the two MOFs. However, in Fig.1a and 1b the characteristic hexagonal facets of M68N are challenging to identify. Could you provide a clear picture, such as 3D reconstruction TEM? Moreover, element mapping images don't provide evidence for the optimal M68N@In-TCPP heterostructure (lines 115-116).

Response: We appreciate the reviewer's comment, which helps us clarify the structure morphological evolution, and formation mechanism of the interfacial boundary within the M68N@In-TCPP heterostructure. We have tried to carry out rotating TEM test for 3D reconstruction TEM images to determine the growth morphology evolution and the exposed facets. It is well-known that MOF is metastable under high-energy electron beam (300 kV) irradiation. Unfortunately, the current authors are not able to collect enough images to access 3D reconstruction TEM imaging of M68N@In-TCPP.

We have identified the characteristic hexagonal facets of pure M68N and M68N@In-TCPP heterostructure via conventional TEM investigation. For easy understanding, we would like to summarize the related data and discussions below.

1) The characteristic hexagonal facets phase identification has been carried out on the heterostructure by X-ray diffraction pattern. It indicates both M68N and In-TCPP are orthogonal structures.

2) TEM bright field images of an M68N nanorod with the exposed characteristic hexagonal facets (100) and (140), with the projected facets parallel to the growth direction (Supplementary Figs. 53a-c). The corresponding electronic diffraction patterns (EDPs) for these facets were designated as [100] and [-22, 5, 16] crystallographic orientations, as seen in Supplementary Figs. 53b-d. A composite stereographic projection centred at [001] in Supplementary Fig. 53e illustrates the alignment parallel to the common axis between the two parallel planes.

Supplementary Fig. 53. Phase identification and growth direction determination using TEM. (a) TEM Bright field image of an individual M68N nanorod with the annotation of one of the exposed facets (100) which is parallel to the growth direction. (b) The EDP obtained at [100] direction. (c) TEM bright field image of another M68N nanorod taken along [-22, 5, 16], around 2.3° away from [-413]. The exposed facet projected trace (140) is also parallel to the growth direction. (d) EDP obtained near zone axis direction [-22,5,16]. It is useful for indexing the projected facet (140). (e) A composited stereographic projection centered at [001]. The red spots denote orientation and the blue ones plane. The growth direction [001] is parallel to the common axis between the two parallel planes (100) and (140).

3) The characteristic exposed facets of M68N in M68N@In-TCPP heterostructure was determined by high-resolution TEM (HRTEM) and Fourier transformation (FFT) in Supplementary Fig. 54. Both the exposed facets of M68N nanorod and In-TCPP nanoflakes are clearly visible in Supplementary Fig. 54. Two specific regions within the interface, marked with hollow red squares are further magnified in Supplementary Figs. 54b and 54d. The lattice fringes in these regions were converted into fast Fourier transform (FFT) patterns, which are then used for diffraction indexing. In Supplementary Figs. 54c and 54e, the two FFT patterns derived from the lattice fringes are assigned to the zone axes [510] and [100], respectively. The exposed facets of the nanoflake are indexed as (021) and (0-21), indicating the nanoflakes project as a square.

Supplementary Fig. 54. Determination of the morphology of In-TCPP mixed with M68N. (a) HRTEM image near the interface area, left side of the M68N nanorod, and the right side In-TCPP nanoflakes. The interface is not sharp due to large size of M68N. The two ROIs in red square are zoomed in and shown in (b) and (d), which are transformed into FFT in (c) and (e). The FFT images are indexed as the diffraction along [510] and [100].

Comment 2. The synthesis of the MOF heterostructure is described in Figure 1c, why use H₂TCPP instead of In-TCPP? (lines 590-591). In addition, is it possible for the trivalent indium at the center of the porphyrin to coordinate axially? Please provide additional evidence to support it. The details of the ¹HNMR in Supplementary Fig. 11 are unclear, and the calculation detail for porphyrin metalation is difficult to understand (Supplementary lines 220-225).

Response: Thank the reviewer for your insightful comments and for recognizing the significance of our work. We have addressed each of your concerns below:

1) Use of H₂TCPP instead of In-TCPP for axial coordination of trivalent indium at the center of porphyrin:

With the addition of tetrakis(4-carboxyphenyl) porphyrin (H₂TCPP), the formed In-TCPP MOF has two chemical states of In, including In-carboxylic (InO₄(OH)₂) and metalloporphyrin (In-N) sites. In the presence of carboxylic group, attempts to pre-metalate TCPP linkers with In³⁺ under the same conditions were unsuccessful because the kinetics of coordinating with carboxyl is much faster than that with porphyrin center. Consequently, we obtained the In-N sites through an in-situ reaction with slow kinetics (Supplementary Fig. 11). In this case, trivalent indium coordinates axially with the nitrogen atoms at the porphyrin centre to form In-N bonds accompanying with InO₄(OH)₂ nodes in MOF structure. This coordination has been confirmed by multiple studies (*J. Am. Chem. Soc.* 2014, 136, 15881; *ACS Catal.* 2018, 8, 4583).

2) Additional evidence to support porphyrin metalation:

We verified the extent of metalation using a combination of NMR spectroscopy, XPS, and ICP-MS. The coordinated In-N percentage is determined by the peak area ratio between the porphyrin's H-1 and the metalated porphyrin's H-1'. Additionally, In 3d spectra confirm the +3 valence state of In-N (Supplementary Fig. 12a). In pure TCPP (2H), peaks at 397.8 eV and 400.1 eV correspond to uncoordinated N and porphyrin center pyrrolic N, respectively. In M68N@In-TCPP, a new peak at 398.9 eV indicates the presence of In-N bonds, with intensity increasing as the content of TCPP(In-N) rises (Supplementary Fig. 12b). ICP-MS results further support the correlation between indium concentration and its mass content in M68N@InTCPP (Supplementary Table 2).

Supplementary Fig. 12. The XPS spectra of M68N@In-TCPP with different content of indium-metallated porphyrin (30 %- 75 %), (a) In 3d, (b) N 1s.

3) Calculation detail for porphyrin metalation based on ¹H NMR spectra:

We have supplemented a series of composites with varied In:TCPP ratio for ¹H NMR measurement in Supplementary Fig. 11. The spectra of acid-digested M68N@In-TCPP exhibit two sets of signals corresponding to free-base porphyrin (TCPP) and indium-metallated porphyrin (TCPP(In)). The peak at 8.61 ppm represents the eight hydrogen atoms of the pyrrole in the TCPP linker (H-1). In the ¹H NMR spectra, the doublet peaks at 8.75 and 8.57 ppm correspond to the outer aromatic hydrogens (H-2) (*Nat. Energy* 2023, 8, 361; *ACS Catal.* 2018, 8, 4583). The percentage of metallated TCPP(In) was determined by integrating the β -pyrrole peaks of metallated porphyrin (H-1', 9.0 ppm, s, 8H) and unmetalled porphyrin (H-1, 8.75 ppm, s, 8H) using the following equation (*J. Am. Chem. Soc.* 2014, 136, 15881; *ACS Catal.* 2018, 8, 4583):

$$C_{\text{In-N}} \% = S_{\text{In-N-9.0}} / (S_{\text{In-N-9.0}} + S_{\text{2H-8.75}}) \times 100 \%$$

Where C is the percentage of metallated TCPP(In); S is the area of β -pyrrole peak in porphyrin.

By adjusting the metal-to-linker ratio, the In-N content can be varied from 30% to 70%. ICP-MS results show a strong correlation between the indium concentration in the porphyrin ring and its mass content in M68N@InTCPP (Supplementary Table 2). The percentage of TCPP(In) was calculated to give a metalation ratio of 47% for the optimal M68N@InTCPP heterostructure.

Supplementary Fig. 11. ^1H NMR spectra of M68N@In-TCPP composites with percentage of metallated TCPP(In) in In-TCPP. The location of the H atom is labeled in the molecular structure of the TCPP and TCPP(In). The values of 30~75% are the percentage of metallated TCPP(In).

Supplementary Table 2. ICP-MS analysis for M68N@In-TCPP.

Sample	In content (wt %)
M68N	27.67
In-TCPP(In-N-30%)	23.63
M68N@In-TCPP-(In-N-30%) ^a	27.20
M68N@In-TCPP-(In-N-47%)	27.55
M68N@In-TCPP-(In-N-60%)	27.79
M68N@In-TCPP-(In-N-75%)	28.05

Note: ^a30 % is the percentage of metallated TCPP(In) in In-TCPP.

We have clarified the calculation details for porphyrin metalation in the caption of Supplementary Fig. 11 on Page 14 line 203: “The disappearance of N-H signal upon coordination with In implies porphyrin ring coordination with In. A detailed comparison of ^1H NMR spectra between TCPP-based MOFs and TCPP alone is presented in Supplementary Fig. 11. The spectra of acid-digested M68N@In-TCPP exhibit two sets of signals corresponding to free-base porphyrin (TCPP) and indium-metallated porphyrin (TCPP(In)). The peak at 8.61 ppm represents the eight hydrogen atoms of the pyrrole in the TCPP linker (H-1). In the ^1H NMR spectra, the doublet peaks at 8.75 and 8.57 ppm correspond to the outer aromatic hydrogens (H-2)^{8,9}. The percentage of metallated TCPP(In) was determined by integrating the β -pyrrole peaks of metallated porphyrin (H-1', 9.0 ppm, s, 8H) and unmetallated porphyrin (H-1, 8.75 ppm, s, 8H) using the following equation^{9,10}:

$$C_{\text{In-N}} \% = S_{\text{In-N-9.0}} / (S_{\text{In-N-9.0}} + S_{\text{2H-8.73}}) \times 100 \%$$

where C is the percentage of metallated TCPP(In); S is the area of β -pyrrole peak in porphyrin.

By adjusting the metal-to-linker ratio, the In-N content can be varied from 30% to 70%. ICP-MS results show a strong correlation between the indium concentration in the porphyrin ring and its mass content in M68N@InTCPP (Supplementary Table 2).

We hope these revisions adequately address your concerns and enhance the clarity of our manuscript. Thank you for your valuable feedback.

Comment 3. While the shift to lower binding energy in the In 3d spectra suggests a reduction in the average valence state of indium, it would be beneficial to provide a more detailed explanation of the underlying mechanisms contributing to this shift. Specifically, elucidating the factors that lead to the transition of indium species from a +3 state to a lower oxidation state would enhance the understanding of the electronic environment in the heterostructure.

Response: Thank you for your valuable suggestion. We have expanded our explanation to clarify the mechanisms underlying the observed shift to lower binding energy in the In 3d spectra, which indicates a reduction in the average valence state of indium.

In our study, we synthesized the M68N@In-TCPP heterostructure using a mixed linker strategy. This approach is widely used to create defect-rich metal nodes, such as Ce(II)-O_v and Zr(III)-O_v (*J. Am. Chem. Soc.* 2016, 138, 6636; *J. Am. Chem. Soc.* 2023, 145, 31, 17164). Initially, the indium nodes (InO₄(OH)₂) in M68N, In-TCPP, and the In-N sites within the porphyrin centre possess a +3 valence state, as shown in Figure 1m and Supplementary Fig. 12a.

During synthesis, the competitive reaction between the two ligands leads to the formation of defected indium nodes. This increases the ratio of In(II) to the total indium (In(III) + In(II)), thereby reducing the average oxidation state of indium to between +2 and +3. Time-dependent X-ray photoelectron spectroscopy (XPS) analysis confirmed that the concentration of defected indium nodes increases as In-TCPP envelops the surface of M68N. Concurrently, the oxygen vacancy content remains at 0% after 0.5 hours but rises from 7.3% to 29.7% as the reaction time extends from 1 to 5 hours. Transmission electron microscopy (TEM) images in Figure 1f-g further support this growth mechanism.

We have updated the manuscript to include the following description on Page 6 line 153: “The X-ray photoelectron spectroscopy (XPS) of In 3d shows a shift to lower binding energies in M68N@In-TCPP, indicating an average indium valence state below +3 ($\text{In}^{\sigma+}$, $2 < \sigma < 3$) (Fig. 1m)^{30,31,37}. Initially, both the indium nodes and In-N sites exhibit a +3 valence state (Figure 1m and Supplementary Fig. 12). Due to the faster nucleation kinetics of the core M68N compared to In-TCPP, indium remains in the +3 oxidation state during the first 0.5 hours before gradually shifting to lower binding energies. The TCPP linker competitively coordinates with unsaturated surface indium nodes (In(III)-O-In(II)), increasing the In(II)/(In(III) + In(II)) ratio and resulting in an average indium oxidation state between +2 and +3. When the TCPP (In-N) content is further

increased to 75%, the In 3d spectra shift back close to the +3 state (Supplementary Fig. 12 and Table 2). The main peak at 531.5 eV in the O 1s XPS spectra of M68N and In-TCPP corresponds to the C=O of the carboxyl groups bonded to indium nodes (Fig. 1n).”

Comment 4. The reported increase in the molar ratio of indium to oxygen atoms from 0.16 to 0.38 is noteworthy. XPS measures the composition of the surface of the material, In the node on the surface of the heterogeneous MOF, the ratio of indium to oxygen may theoretically be less than M68N (0.19), but it should not be greater than In-TCPP (0.20). It would strengthen the manuscript to include a discussion on the significance of this increase. For example, how does this change affect the structural integrity and functionality of the heterostructure? The spatial distribution of the MOF heterogeneous interface can be deepened by surface etching and the change of In-O ratio measured by XPS.

Response: Thank you for your valuable suggestion. We apologise for the unclear description in our initial manuscript.

In the self-assembly process of In-TCPP, the kinetics of In coordinating with carboxyl is much faster than that with porphyrin center. Consequently, most of In atoms were coordinated with carboxylic groups of TCPP to form a bulk stacking MOF structure due to strong π - π stacking interaction, while a little part of In atoms were coordinated with porphyrin center to yield only 30% of metalation porphyrin with the coexistence of more uncoordinated pyrrolic N atoms. In the presence of two types of organic linkers, the difference in growth kinetics between the two MOFs enabled the envelopment of M68N by In-TCPP nanosheets. Thus we obtained more exposed In-N sites through an in-situ reaction with slow kinetics (Supplementary Fig. 11). In return, the lattice oxygen vacancies of In-O (O_v) become more pronounced (Fig. 1n). With extended reaction time, the molar ratio of indium to oxygen atoms on the surface increased from 0.16 to 0.38 (Supplementary Table 3), indicating the gradual formation of In-TCPP nanosheets enveloping M68N core.

We have incorporated the following description on Page 6 line 160 in the revised manuscript: “The X-ray photoelectron spectrometer (XPS) spectra of In 3d revealed a shift to lower binding energy in M68N@In-TCPP, indicating an average valence state of indium below +3 ($In^{\sigma+}$, $2 < \sigma < 3$) (Fig. 1m)^{30,31,37}. In single-component MOFs, indium mainly exists as +3 when coordinating with each organic linker. During the self-assembly process of In-TCPP, the kinetics of In coordinating with carboxyl is much faster than that with porphyrin center. Consequently, most of In atoms were coordinated with carboxylic groups of TCPP to form a bulk stacking MOF structure due to strong π - π stacking interaction, while a little part of In atoms were coordinated with porphyrin center to yield only 30% of metalation porphyrin with the coexistence of more uncoordinated pyrrolic N atoms (Supplementary Fig. 11). During the formation of MOF heterostructures, indium species exist in a +3 oxidation state within the first 0.5 h and then gradually shift to a lower binding energy. The main peak at 531.5 eV in O 1s XPS spectra of M68N and In-TCPP is assigned to the

C=O of the carboxyl that connected with In nodes (Fig. 1n)²⁸. The O 1s peak shifts to higher binding energy (531.8 eV) with extended reaction time. **The difference in growth kinetics between the two MOFs enabled the envelopment of M68N by In-TCPP nanosheets with more exposed In sites due to their slow growth kinetics.** In return, a new shoulder peak at around 529.9 eV, corresponding to the lattice oxygen vacancies of In-O (O_v), becomes more pronounced. With extended reaction time, the molar ratio of indium to oxygen atoms on the surface increased from 0.16 to 0.38 (Supplementary Table 3), **indicating the gradual formation of In-TCPP nanosheets enveloping M68N core.** These results demonstrate that the formation of defective indium sites with oxygen vacancies is essential for the heterostructure formation, arising from the unsaturated indium nodes at the interface between the two MOFs and In-TCPP nanosheets.”

Comment 5. The expression about the effect of hydrophilicity on the H₂O₂ performance in lines 305-308 is contradictory with the actual performance results.

Response: Thank you for raising this insightful question. To address the spatial distribution of the core-shell M68N@In-TCPP heterostructure, we performed hydrophilicity testing. The water contact angles of the individual components reveal that M68N is hydrophilic (26°), while In-TCPP is mildly hydrophobic (90°). The composite M68N@In-TCPP exhibited a contact angle of 82°, confirming that the outer shell is In-TCPP, while the inner core is M68N.

The hydrophilicity of M68N supports the oxidation of H₂O to H₂O₂. However, M68N alone demonstrates poor activity for H₂O₂ production due to its limited electron-hole separation efficiency. The superior performance of M68N@In-TCPP, compared to M68N alone, arises from the heterojunctions formed between M68N and In-TCPP. These heterojunctions enhance the efficient transfer of photogenerated charges, enabling H₂O₂ production at the M68N sites and CO₂ reduction to HCOOH at the In-TCPP sites, as discussed in the manuscript. We believe this explanation provides clarity on the spatial distribution and functional advantages of the core-shell structure in the composite.

Comment 6. In general, the yield of photocatalysis under natural light is lower than that achieved with xenon lamp irradiation. However, when comparing natural light to visible light irradiation in this manuscript, the yield under natural light can be twice as high (Figure 2a and 2b). Please give the possible discrepancy factors.

Response: Thank you for your thoughtful comment. We agree that the higher yield observed under focused natural sunlight compared to xenon lamp irradiation warrants further explanation. Two primary factors contribute to this discrepancy:

1) Light Intensity: The intensity of focused natural sunlight is approximately 1.3 W cm⁻², which is nearly 4.5 times greater than the xenon lamp's intensity of 0.3 W cm⁻². This higher intensity generates significantly more photo-induced electrons, enhancing the photocatalytic performance under sunlight.

2) Ultraviolet (UV) Light Component: Natural sunlight contains a UV component, which contributes to higher photocatalytic yields (see Figure 2i). In contrast, the xenon lamp, equipped with a 400 nm cutoff filter, emits only visible light and excludes UV irradiation. As a result, the yield under focused natural sunlight is approximately twice as high as that under the xenon lamp.

To address potential discrepancies more comprehensively, we conducted additional tests under unfocused natural sunlight (75 mW cm^{-2}). These results showed similar photocatalytic performance compared to the xenon lamp with an AM 1.5G filter, demonstrating the reliability of the sunlight-driven system (Supplementary Fig. 15).

Supplementary Fig. 15. The photocatalytic performance of M68N@In-TCPP using different light sources: sunlight irradiation with a light intensity of 75 mW cm^{-2} and AM 1.5 G (75 mW cm^{-2})

We have updated the manuscript accordingly on Page 8 line 204: “The H₂O₂ yield was slightly lower at $321.2 \mu\text{mol g}^{-1} \text{h}^{-1}$ compared to the HCOOH yield, likely due to its decomposition under intense light conditions. Additional tests under unfocused natural sunlight (75 mW cm^{-2}) showed comparable performance to the xenon lamp with an AM 1.5G filter, confirming the reliability of the sunlight-driven photocatalysis system (Supplementary Fig. 15). Despite fluctuations in light intensity ($1.0\text{--}1.3 \text{ W cm}^{-2}$), the average $\eta\%$ value was calculated to be 0.04%, surpassing the performance of most recently reported CO₂ photoreduction systems (Supplementary Table 4).”

We believe this explanation addresses the observed discrepancies and highlights the robustness of the photocatalytic system under natural sunlight.

Comment 7. For the overall photosynthesis, the oxidation and reduction products should have a certain molar ratio according to the reaction equation. In this manuscript, the total number of moles for oxidation products (H₂O₂ and O₂) shown in Figure 2b and 2g is less than the theory, please analyze the reason. In addition, whether the quantification of the product requires multiple tests and displays of error rods.

Response: Thank the reviewer for your insightful comment. We have carefully analysed the discrepancy in the molar ratio of oxidation and reduction products and addressed the need for

error bars in product quantification.

In Figs. 2b and 2g, the calculated electron-to-hole (e^-/h^+) ratio is 1.09, slightly higher than the theoretical value of 1. This minor difference can be attributed to the formation of undetected reactive oxygen species, as supported by similar observations in the literature (*Nat. Commun.* 2022, 13, 4592). Additionally, the decomposition of H_2O_2 , occurring at a rate of $4 \mu\text{mol g}^{-1} \text{h}^{-1}$, contributes to the observed discrepancy in hole consumption (**Supplementary Fig. 24**). We have included a comparison of e^-/h^+ values reported in related studies on H_2O_2 photosynthesis from CO_2 in Table R1 for reference.

To ensure clarity and accuracy, we have conducted multiple tests to quantify the products and have now included error bars in all relevant figures. We believe this analysis clarifies the observed discrepancies and strengthens the discussion of the product quantification process.

Supplementary Fig. 24. The decomposition test behaviours toward H_2O_2 with the M68N@In-TCPP under light irradiation.

Table R1. Comparison with the ratio of total consumption electron number and total consumption hole number based on the photocatalytic CO_2 overall reaction.

Photocatalyst	Reduction Product ($\mu\text{mol g}^{-1} \text{h}^{-1}$)	Oxidation product ($\mu\text{mol g}^{-1} \text{h}^{-1}$)	TCEN ^a ($\mu\text{mol g}^{-1} \text{h}^{-1}$)	TCHN ^b ($\mu\text{mol g}^{-1} \text{h}^{-1}$)	TCEN/T CHN	Ref
PCN-601	CH_4 (10.1) CO (5.3)	H_2O_2 (37.5)	91.6	75	1.22	J. Am. Chem. Soc. 2020, 142, 12515.
$Ag_{27}TPyP-Fe$	CO (36.5)	H_2O_2 (35.9)	73	71.8	1.02	Angew. Chem. Int. Ed. 2024, e202412553.
TMOF-10- NH_2 (I)	CO (78)	H_2O_2 (36.3)	156	72.6	2.15	Nat Commun. 2022, 13, 4592.
$FeTCP-OH-Co$	CO (0.4) HCOOH (17.3)	H_2O_2 (12.8)	35.4	25.6	1.38	Angew. Chem. Int. Ed. 2022, 61, e202111622.
$WO_3 \cdot 0.33.H_2O$	HCOOH (1.5) CH_3COOH (9.8)	H_2O_2 (35.2)	81.4	70.4	1.15	J. Am. Chem. Soc. 2018, 140,

M68N@In-	CO (22.4)	O ₂ (5.4)	287.2	260.2	1.1	This work
TCPP	HCOOH(121.1)	H ₂ O ₂ (119.3)				

Note: ^aTECN: Total consumption electron number.

^bTCHN: Total consumption hole number.

Fig. 2. High performance of sunlight-driven photoreduction CO₂ and coupling H₂O oxidation. (a) Practical performance test and corresponding STC conversion efficiency of M68N@In-TCPP photocatalyst under 4 h of natural sunlight irradiation between January 10 and 13th, 2024. (b) Product yields over different photocatalysts under visible light ($\lambda \geq 400$ nm). (c) GC-MS result of ¹³CO and (d) ¹³C NMR spectra of H¹³COOH from ¹³CO₂ isotopic experiment over M68N@In-TCPP. (e) Photocatalytic performance of M68N@In-TCPP in diluted CO₂ (10 vol% CO₂ in Ar). (f) Long-term measurement of HCOOH and H₂O₂ production from overall photocatalytic CO₂ reduction. (g) Products from photocatalytic reaction during 5-cycle tests. (h) UV-vis DRS spectra of different samples. (i) AQY values under various wavelengths of light irradiation. The overlap is the spectrum of solar irradiation.

The revised manuscript includes the following description on Page 10 line 268: “This indicates that H₂O oxidation is closely linked to electron consumption by CO₂ at the reductive

sites of In-TCPP. The e^-/h^+ ratio in Figure 2b is 1.09, which slightly deviates from the theoretical value due to the formation of undetected reactive oxygen species and the decomposition of H_2O_2 (Supplementary Fig. 24). This observation aligns with many reported works¹⁶. Regarding CO_2 reduction, In-TCPP exhibited high selectivity for HCOOH production (81%), closely matching that of the heterostructure and surpassing M68N (65%).”

Comment 8. For in-situ DRIFTS spectra of M68N and M68N@In-TCPP in Figure 3g and 3h, what might be the reason for the big difference in the signal-to-noise ratio? And DRIFTS peak is not obvious, whether there is a way to measure more convincing signal strength.

Response: Thank you for your valuable feedback. To address the differences in signal-to-noise ratios observed in the in-situ DRIFTS spectra of M68N and M68N@In-TCPP (Figure 3g and 3h), we optimize the operation conditions of DRIFTS measurement to enhance signal strength. The improved signal quality is presented in the revised Figure 3f–i, and our analysis is detailed below.

Figure 3. In situ DRIFTS spectra from 1300 to 1800 cm^{-1} . Reactants adsorption on (f) M68N and (g) M68N@InTCPP in the dark. Intermediates evolution on (h) M68N and (i) M68N@InTCPP during photocatalytic reaction.

Supplementary Fig. 45. In situ DRIFTS tests in the region of 3400-3800 cm^{-1} on (a) M68N and (b) M68N@In-TCPP in the dark.

Comment 9. The proposed two pathways for CO_2 and H_2O adsorption are intriguing. It would enhance clarity to discuss the physical basis for the preference of CO_2 adsorption at the In-N sites over $\text{InO}_4(\text{OH})_2$ sites in more detail. Additionally, including energy diagrams or transition state structures could help visualize the differences in activation energies and elucidate the underlying mechanisms of the adsorption processes.

Response: Thank the reviewer for your valuable suggestion. To clarify the preference of CO_2 adsorption at In-N sites over $\text{InO}_4(\text{OH})_2$ sites, we have supplemented the calculations on the parameters of CO_2 adsorption states and energy diagrams.

We evaluated multiple adsorption sites to determine the most stable configurations with the lowest adsorption energies, as illustrated in Figs. 3b–c and Supplementary Figs. 41–42. For CO_2 adsorption on $\text{InO}_4(\text{OH})_2$ sites, the adsorption energy was found to be -0.056 eV with an In-O distance of 3.38 Å (Supplementary Fig. 42). In contrast, CO_2 adsorption at In-N sites showed the adsorption energy of -0.09 eV with a shorter In-O distance of 2.90 Å. The comparison in the CO_2 adsorption state supports the preferential adsorption of CO_2 molecules at In-N sites, highlighting a stronger bonding interaction compared to $\text{InO}_4(\text{OH})_2$ sites.

Supplementary Fig. 42. DFT calculations of the distance between the adsorbed CO_2 and In atoms in (a) $\text{InO}_4(\text{OH})_2$ and In-N (b) sites.

To elucidate the underlying mechanisms, we further constructed free energy diagrams (FED) to investigate the reduction pathway of CO₂ to HCOOH at both InO₄(OH)₂ and In-N sites. The conversion process involves two hydrogenation steps: from *CO₂ to HCOO* and from HCOO* to HCOOH*, with the latter being the rate-determining step. As shown in Fig. 3j, the energy barrier for the RDS at In-N sites is 0.53 eV, significantly lower than the 0.83 eV barrier at InO₄(OH)₂ sites. This lower energy barrier at In-N sites indicates a more favourable pathway for HCOOH formation, thereby establishing In-N as the active site for CO₂ photoreduction.

Figure 3j. Calculated Gibbs free energy profile for the production of HCOOH on In-N and In₂O₄(OH)₂ sites of M68N@In-TCPP.

These enhancements provide a clearer physical basis for the preference of CO₂ adsorption at In-N sites and offer energy diagrams to understand the reaction pathways. We believe these revisions significantly improve the clarity and depth of our discussion regarding the adsorption mechanisms within the heterostructure.

In response to your suggestion, we have revised the manuscript to include a detailed discussion of these findings. The revised sections now elaborate on the adsorption energies, bond lengths, electron density redistribution, and the free energy diagrams that illustrate the differences in activation energies between the two adsorption sites. Specifically, the manuscript now includes:

Page 13 line 384: “Density function theory (DFT) calculations provided a comprehensive analysis of the adsorption performance of CO₂ and H₂O on In₂O₄(OH)₂ and In-N sites, including optimal adsorption energies and bond lengths (Figs. 3b and c, Supplementary Figs. 41 and 42). Two theoretical pathways were proposed for reactant adsorption: one where CO₂ adsorbs at the In-N site and H₂O at InO₄(OH)₂ sites (Fig. 3b-c), and another with reversed adsorption (Supplementary Fig. 41). These calculations suggest that H₂O is more readily activated at InO₄(OH)₂ sites. For CO₂ adsorption on InO₄(OH)₂ sites, the adsorption energy was found to be -0.056 eV with an In-O distance of 3.38 Å (Supplementary Fig. 42). In contrast, CO₂ adsorption at In-N sites showed the adsorption energy of -0.09 eV with a shorter In-O distance of 2.90 Å. The comparison in the CO₂ adsorption state supports the preferential adsorption of CO₂ molecules at

In-N sites, highlighting a stronger bonding interaction compared to $\text{InO}_4(\text{OH})_2$ sites. Electron density redistribution calculations showed CO_2 molecules gaining 0.006 electrons from the In-N dimers, while H_2O molecules lost 0.057 electrons to $\text{InO}_4(\text{OH})_2$, facilitating photoinduced electrons transferred from In-TCPP to CO_2 at In-N sites and from the adsorbed H_2O to $\text{InO}_4(\text{OH})_2$ sites⁵².”

Page 15 line 446: “Thus, the M68N@In-TCPP heterostructure enhances the adsorption and conversion of CO_2 and H_2O through plausible pathways of $\text{CO}_2 \rightarrow \text{CO}_2^- \rightarrow \text{HCOO}^* \rightarrow \text{HCOOH}$ and $\text{H}_2\text{O} \rightarrow \bullet\text{OH} \rightarrow \text{H}_2\text{O}_2$ ^{16,60,61}. Additionally, we have calculated the Gibbs free energy of reaction intermediates to plot free energy diagrams (FED) to elucidate the underlying CO_2 reduction pathway. The conversion process involves two hydrogenation steps: from $^*\text{CO}_2$ to HCOO^* and from HCOO^* to HCOOH^* , with the latter being the rate-determining step (RDS). As shown in Fig. 3j, the energy barrier for the RDS at In-N sites is 0.53 eV, significantly lower than the 0.83 eV barrier at $\text{InO}_4(\text{OH})_2$ sites. This lower energy barrier at In-N sites indicates a more favourable pathway for HCOOH formation, thereby establishing In-N as the active site for CO_2 photoreduction.”

Manuscript NCOMMS-24-54153-A Response to Reviewers' Comments

Given below are our responses (in BLUE colour) to the reviewer's comments. The changes to the manuscript and supplementary information are marked in RED colour.

Reviewer #2 (Remarks to the Author):

Comment. The authors addressed some of my concerns. However, the major concerning point remain unclear, which should be further clarified

Response: We appreciate the reviewer's recognition on the improvement of manuscript quality. We also thank the reviewer for the additional suggestions and have carefully addressed the reviewer's comments point-wise in our response below.

Comment 1. Although the authors determined the HOMO and LUMO levels of each component, the HOMO of M68N (1.8 V) is not high enough to oxidize water to produce OH radicals (> 1.9 V).

Response: We appreciate the reviewer's suggestion to help us clarify the •OH generation. The HOMO of M68N (1.8 V vs NHE) is not high enough to oxidize water to produce •OH radicals due to the high energy requirement of oxidizing H₂O to •OH radicals ($\text{H}_2\text{O} + \text{h}^+ = \bullet\text{OH} + \text{H}^+$, 1.97 V vs NHE). The •OH radicals can be generated by the reduction of H₂O₂ on photocatalyst surface ($\text{H}_2\text{O}_2 + \text{e}^- = \bullet\text{OH} + \text{OH}^-$, 0.73 V vs NHE) (*Nat. Chem.* 2024, 16, 1250; *Chem. Rev.* 2017, 117, 11302; *Coord. Chem. Rev.* 2016, 143, 327). EPR spectrum in Supplementary Fig. 46 suggests the generation of •OH radicals accompanying by H₂O₂ formation.

In response to the reviewer's suggestion, we have clarified the •OH formation on Page 15 line 443 of the manuscript: "During the photocatalytic reaction on M68N@In-TCPP, a distinct 1:2:2:1 quadruplet peak in EPR spectrum is attributed to the DMPO-•OH adducts (Supplementary Fig. 46). Since the HOMO of M68N (1.8 V vs NHE) is not high enough to oxidize water to produce •OH radicals ($\text{H}_2\text{O} + \text{h}^+ = \bullet\text{OH} + \text{H}^+$, 1.97 V vs NHE), the •OH radicals can be generated by the reduction of H₂O₂ on photocatalyst surface ($\text{H}_2\text{O}_2 + \text{e}^- = \bullet\text{OH} + \text{OH}^-$, 0.73 V vs NHE)⁶²⁻⁶⁴."

Comment 2. How can the organic material keep stable in the presence of produced OH radicals? Furthermore, OH radicals can also react with H₂O₂ at a high reaction rate.

Response: Thank you for your comment. Even though H₂O₂ is a strong two-electron oxidant, it exhibits poor or no reactivity to decompose most organic molecules due to a high activation energy barrier. The activation of H₂O₂ can be improved by facilitating one-electron reduction to •OH. However, this activation process is not efficient, leading to the low H₂O₂ utilization. That is why most literatures introduced metal ions or nanoparticles to assist in the H₂O₂ activation to enhance the H₂O₂ utilization (*Angew. Chem. Int. Ed.* 2022, 61, e202200670; *Angew. Chem. Int.*

Ed. 2020, 59, 16517; *Coord. Chem. Rev.* 2016, 143, 327). If the valence band is incapable to oxidize H₂O to •OH, the dominant active species for organic molecule degradation are normally reported to be •O₂⁻ or ¹O₂ by O₂ activation rather than •OH radicals by H₂O₂ reduction (*J. Am. Chem. Soc.* 2023, 145, 16852; *Adv. Funct. Mater.* 2024, 2419010; *Chem. Eng. J.* 2020, 382, 123017). In CO₂ atmosphere, the reduction of O₂ to •O₂⁻ is inhibited, so the degradation of organic materials can be suppressed. Moreover, the combination of high-valent metals (Zr⁴⁺, Fe³⁺, In³⁺, and Al³⁺) and carboxylate (COO⁻) based linker (TCPP or NH₂-BDC) can form a strong coordination bond according to the hard–soft–acid–base (HSAB) principle (*J. Am. Chem. Soc.* 2023, 145, 18931; 15881. *ACS Catal.* 2018, 8, 5, 4583). Therefore, the M68N@In-TCPP can remain stable in the presence of low H₂O₂ and •OH level during the reaction. The evidences in the stable structure of M68N@In-TCPP have been presented in Supplementary Figs. 32-37 and Table 6.

The produced •OH can reacts with H₂O₂ via the equation (H₂O₂ + •OH = •O₂⁻ + H₂O + H⁺) (*Angew. Chem. Int. Ed.* 2022, 61, e202200670; *Chem. Rev.* 2017, 117, 11302). The produced •O₂⁻ may also react with H₂O₂, which is known as the Haber-Weiss reaction (H₂O₂ + •O₂⁻ = •OH + O₂ + OH⁻) (*Chem. Rev.* 2017, 117, 11302). The quenching of H₂O₂ with •OH can be observed in the presence of high H₂O₂ dosage (≥ 1.0 mM) (*Angew. Chem. Int. Ed.* 2020, 59, 16517). In our study, the production of H₂O₂ is only 0.3 mM after 5 h of reaction. In addition, the rapid release of H₂O₂ from catalyst surface into the solution will avoid the H₂O₂ consumption (*Angew. Chem. Int. Ed.* 2022, 61, e202200670). Therefore, the reaction of H₂O₂ with •OH will be suppressed to keep H₂O₂ as the stable reactive oxygen species.

In response to the reviewer's suggestion, we have added the related explanations on Page 15 line 443 of the manuscript: "During the photocatalytic reaction on M68N@In-TCPP, a distinct 1:2:2:1 quadruplet peak in EPR spectrum is attributed to the DMPO–•OH adducts (Supplementary Fig. 46). Since the HOMO of M68N (1.8 V vs NHE) is not high enough to oxidize water to produce •OH radicals (H₂O + h⁺ = •OH + H⁺, 1.97 V vs NHE), the •OH radicals can be generated by the reduction of H₂O₂ on photocatalyst surface (H₂O₂ + e⁻ = •OH + OH⁻, 0.73 V vs NHE)⁶²⁻⁶⁴. Even the produced •OH can reacts with H₂O₂ via the equation (H₂O₂ + •OH = •O₂⁻ + H₂O + H⁺), the consumption of H₂O₂ can be avoided due to its low dosage as well as the rapid release into the solution^{65,66}."

Comment 3. Was H₂O₂ produced via single-electron WOR or two-electron WOR? The authors should discuss the mechanism of H₂O₂ formation with more solid evidences.

Response: We appreciate the reviewer's suggestion to help us clarify the water oxidation pathway. The H₂O-to-H₂O₂ oxidation has been proposed to be a direct two-electron oxidation process (2H₂O + 2h⁺ = H₂O₂ + 2H⁺) or two-step single-electron pathway (H₂O + h⁺ = •OH + H⁺; •OH + •OH = H₂O₂). The HOMO of M68N (1.8 V vs NHE) is more negative than the potential of H₂O/•OH (1.97 V vs NHE) but higher than the potential of H₂O/H₂O₂ (1.35 V vs NHE), the H₂O-

to-H₂O₂ oxidation is more likely to proceed in a direct two-electron oxidation process (*Nat. Chem.* 2024, 16, 1250, *Nat Commun.* 2022, 13, 4592).

To explore the mechanism of water oxidation in CO₂ atmosphere, we conducted the photocatalytic experiments with different scavengers. Tert-butanol (TBA) and KBrO₃ were introduced as the scavenger for •OH radicals and electrons, respectively (Supplementary Fig. 47). The addition of TBA slightly decreased the H₂O₂ production (112.2 μmol g⁻¹ h⁻¹), excluding the possibility of an •OH-mediated single-electron pathway for water oxidation. In contrast, the production rate of H₂O₂ was significantly increased from 119.3 to 185.4 μmol g⁻¹ h⁻¹ in the presence of KBrO₃. The accelerated H₂O-to-H₂O₂ oxidation rate can be ascribed to the increased charge separation by rapid consumption of electrons. Further, the change of •OH concentration was monitored by a fluorescence probe of 2-hydroxyterephthalic acid and compare to that of H₂O₂ concentration during the photocatalytic reaction (Supplementary Fig. 48). Different from the linearly increase in H₂O₂ production, the •OH generation increased at the initial 30 min of reaction and gradually became saturated with the prolonged light irradiation. The distinct trend between them indicates that H₂O₂ should not be formed by •OH coupling. Instead, the saturation state of •OH concentration resembles the feature as its formation from H₂O₂ reduction (*Angew. Chem. Int. Ed.* 2022, 61, e202200670). Therefore, it can be concluded that photocatalytic H₂O oxidation in CO₂ generates H₂O₂ over M68N@In-TCPP via a two-electron oxidation process.

Supplementary Fig. 47. The H₂O₂ yield over M68N@In-TCPP in CO₂ atmosphere with different sacrificial agents (10 mM KBrO₃, and TBA). (a) The H₂O₂ yield with reaction time. (b) The corresponding H₂O₂ production rate.

Supplementary Fig. 48. (a) Fluorescence (PL) intensity of the reaction product between •OH radical and fluorescence probe of 2-hydroxy terephthalic acid (TPA) during the photocatalytic reaction. (b) The comparison in the change of •OH level to that of H₂O₂ yield during the photocatalytic reaction.

In response to the reviewer's suggestion, we have added the related explanations on Page 15 line 443 of the manuscript: "During the photocatalytic reaction on M68N@In-TCPP, a distinct 1:2:2:1 quadruplet peak in EPR spectrum is attributed to the DMPO-•OH adducts (Supplementary Fig. 46). Since the HOMO of M68N (1.8 V vs NHE) is not high enough to oxidize water to produce •OH radicals ($\text{H}_2\text{O} + \text{h}^+ = \bullet\text{OH} + \text{H}^+$, 1.97 V vs NHE), the •OH radicals can be generated by the reduction of H₂O₂ on photocatalyst surface ($\text{H}_2\text{O}_2 + \text{e}^- = \bullet\text{OH} + \text{OH}^-$, 0.73 V vs NHE)⁶²⁻⁶⁴. Even the produced •OH can react with H₂O₂ via the equation ($\text{H}_2\text{O}_2 + \bullet\text{OH} = \bullet\text{O}_2^- + \text{H}_2\text{O} + \text{H}^+$), the consumption of H₂O₂ can be avoided due to its low dosage as well as the rapid release into the solution^{65,66}. To explore the mechanism of water oxidation in CO₂ atmosphere, we conducted the photocatalytic experiments with different scavengers. Tert-butanol (TBA) and KBrO₃ were introduced as the scavenger for •OH radicals and electrons, respectively (Supplementary Fig. 47). The addition of TBA slightly decreased the H₂O₂ production (112.2 μmol g⁻¹ h⁻¹), excluding the possibility of an •OH-mediated single-electron pathway for water oxidation. In contrast, the production rate of H₂O₂ was significantly increased from 119.3 to 185.4 μmol g⁻¹ h⁻¹ in the presence of KBrO₃. The accelerated H₂O-to-H₂O₂ oxidation rate can be ascribed to the increased charge separation by rapid consumption of electrons. Further, the change of •OH concentration was monitored by a fluorescence probe of 2-hydroxyterephthalic acid and compared to that of H₂O₂ concentration during the photocatalytic reaction (Supplementary Fig. 48). Different from the linearly increase in H₂O₂ production, the •OH generation increased at the initial 30 min of reaction and gradually became saturated with the prolonged light irradiation. The distinct trend between them indicates that H₂O₂ should not be formed by •OH coupling. Instead, the saturation state of •OH concentration resembles the feature as its formation from H₂O₂ reduction⁶⁵. It can be concluded that photocatalytic H₂O oxidation in CO₂ generates H₂O₂ over M68N@In-TCPP via a two-electron oxidation process. Thus, the M68N@In-TCPP heterostructure enhances the adsorption and conversion of CO₂ and H₂O through plausible pathways of $\text{CO}_2 \rightarrow \text{CO}_2^- \rightarrow$

$\text{HCOO}^* \rightarrow \text{HCOOH}$ and $\text{H}_2\text{O} \rightarrow \text{H}_2\text{O}_2$ ^{16,20}.”

Comment 4. In the ^{13}C isotopic experiment, the signal was very low in comparison with the noise, did authors detect H^{12}COOH and compare it with H^{13}COOH ?

Response: We appreciate the reviewer’s suggestion and have supplemented the ^1H NMR spectrum of isotopic products. The ^{13}C NMR spectrum was repeated to improve the signal-to-noise ratio. In $^{12}\text{CO}_2$ atmosphere, the produced H^{12}COOH displays a singlet peak at 8.30 ppm in ^1H NMR spectrum. When replacing $^{12}\text{CO}_2$ by $^{13}\text{CO}_2$, HCOOH products are detected as doublet peaks at 8.56 and 8.07 ppm arising from ^1H - ^{13}C J -coupling (Fig. 2d), whereas the H^{12}COOH signal is negligible due to the high $^{13}\text{CO}_2$ purity of 99.9% (*Nat. Chem.* 2023, 15, 794; *J. Am. Chem. Soc.* 2022, 144, 3626; *Angew. Chem. Int. Ed.* 2023, 62, e202304050). In addition, ^{13}C NMR spectrum showed a distinct signal for $^{13}\text{HCOOH}$ at 163.4 ppm (Fig. 2e). The results indicate that the formation of HCOOH was originated from CO_2 reduction.

In response to the reviewer’s suggestion, we have added the ^1H NMR spectrum in Fig. 2d and the corresponding description on Page 11 line 285: “Liquid products were analysed by ^1H NMR and ^{13}C NMR. In $^{12}\text{CO}_2$ atmosphere, the produced H^{12}COOH displays a singlet peak at 8.30 ppm in ^1H NMR spectrum. When replacing $^{12}\text{CO}_2$ by $^{13}\text{CO}_2$, HCOOH products are detected as doublet peaks at 8.56 and 8.07 ppm arising from ^1H - ^{13}C J -coupling (Fig. 2d)^{7,46,47}, whereas the H^{12}COOH signal is negligible due to the high $^{13}\text{CO}_2$ purity of 99.9%. In addition, ^{13}C NMR spectrum showed a distinct signal for $^{13}\text{HCOOH}$ at 163.4 ppm (Fig. 2e). The results indicate that the formation of HCOOH was exclusively originated from CO_2 reduction.”

Fig. 2. (d) ^1H NMR and (e) ^{13}C NMR spectra of liquid products from photocatalytic CO_2 reduction using $^{12}\text{CO}_2$ or $^{13}\text{CO}_2$.

Comment 5. Minor: The two subtitles “Photocatalytic CO₂ reduction under natural sunlight” and “Photocatalytic CO₂ reduction and H₂O oxidation” in the “Methods” are confusable.

Response: According to the suggestion, we have updated the two subtitles and distinguished them by marking the light source.

Page 23 line 677:

“Measurement of solar-to-chemical conversion efficiency (outdoor natural sunlight)

M68N@InTCPP photocatalyst was diluted in 15 mL of water and maintained at 25 °C via circulating condensate...

Page 23 line 688:

Visible-light driven photocatalytic experiments (indoor online test equipped with Xe lamp)

The experiments were performed in a 350 mL reactor under 1 atm CO₂ kept at room temperature...”